# Mesoscale weather systems and associated potential wind power variations in a mid-latitude sea strait (Kattegat)

Jérôme Neirynck[1], Jonas Van de Walle[1], Ruben Borgers[1], Sebastiaan Jamaer[1], Johan Meyers[2], Ad Stoffelen[3], and Nicole P.M. van Lipzig[1]

[1]Earth and Environmental Sciences, KU Leuven, Belgium
[2]Mechanical Engineering, KU Leuven, Belgium
[3]R&D Satellite Observations, Royal Netherlands Meteorological Institute, De Bilt, The Netherlands

**Correspondence:** Jérôme Neirynck (jerome.neirynck@kuleuven.be)

**Abstract.** Mesoscale weather systems cause spatiotemporal variability in offshore wind power and insight in their fluctuations can support grid operations. In this study, a 10-year model integration with the kilometre-scale atmospheric model COSMO-CLM served a wind and potential power fluctuation analysis in the Kattegat, a mid-latitude sea strait of 130 km width with an irregular coastline. The model agrees well with scatterometer data away from coasts and small islands, with a spatiotemporal root mean square difference of 1.35 m/s. A comparison of 10 minute wind speed at about 100 metre with lidar data for a 2 year period reveals a very good performance with a slight model overestimation of 0.08 m/s and a high value for the Perkins Skill Score (0.97). From periodograms made using the Welch method it was found that the wind speed variability on a sub-hourly timescale is higher in winter compared to summer. In contrast, the wind power varies more in summer when winds often drop below the rated power threshold. During winter, variability is largest in the northeastern part of the Kattegat due to a spatial spin up of convective systems over the sea during the predominant southwesterly winds. Summer convective systems are found to develop over land, driving spatial variability in offshore winds during this season. On average over the 10 summers the mesoscale wind speeds are up to 20% larger than the synoptic background at 17 UTC with a clear diurnal cycle. The winter averaged mesoscale wind component is up to 10% larger with negligible daily variation. Products with a lower resolution like ERA5 substantially underestimate this ratio between the mesoscale and synoptic wind speed. Moreover, taking into account mesoscale spatial variability is important for correctly representing temporal variability of power production. The root mean square difference between two power output time series, one ignoring and one accounting for mesoscale spatial variability, is 14% of the total power generation.

**Keywords.** Regional climate modelling, mesoscale systems, wind speed variability, wind power fluctuations

## 1 Introduction

According to the sixth iteration of the Intergovernmental Panel on Climate Change (IPCC) assessment report wind energy is one of the capital ways of reducing greenhouse gas emissions (Shukla et al., 2022). Therefore, offshore wind energy will be playing an increasingly important role in our electricity grid. Compared to onshore, offshore locations are advantageous

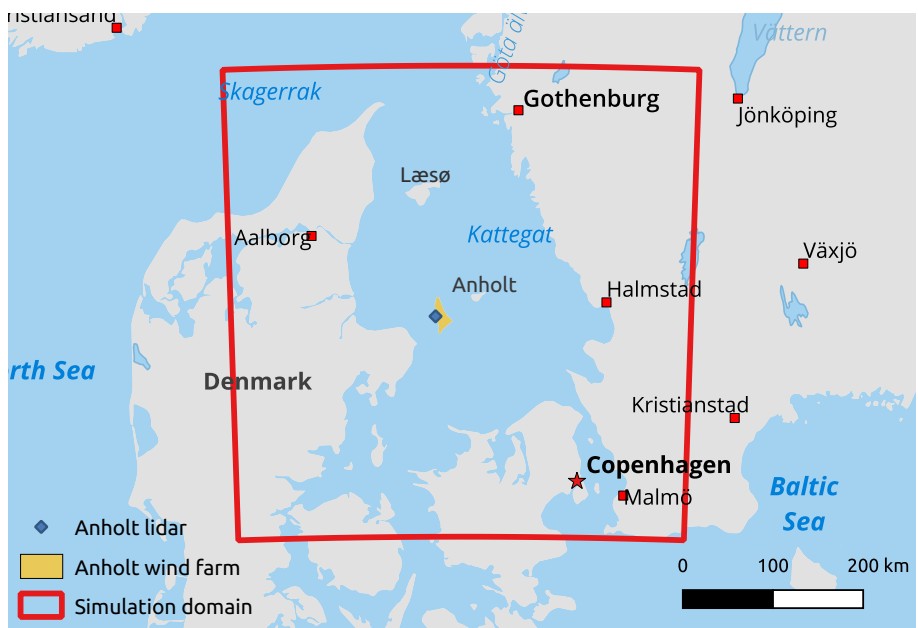

**Figure 1.** Map of the simulation domain. On every side of the domain 20 points are used for relaxation and spin up zones. These relaxation and spin up points are not used in the analysis. Made with Natural Earth.

because the offshore environment generally features greater wind speeds (Kaldellis and Kapsali, 2013). Additionally they also have a higher acceptance in society since offshore wind turbines are mostly placed out of sight (Betakova et al., 2015).

Over the years, individual wind turbines have grown significantly in size and offshore wind project areas are now much bigger than ten years ago, thereby strongly condensing the power generating capacity (Díaz and Soares, 2020). Hence, such condensed wind farms are having a direct impact on the power grid. In addition, in the absence of large energy storage facilities, electricity production always needs to match demand (Hossain and Pota, 2014). A condensed wind farm with low production

during the peak demand hours is detrimental. In brief, with changing wind farm layout and challenging energy demand, wind variations at increasingly fine spatial and temporal scales have to be investigated. As an example, a thunderstorm system passing a condense wind farm during peak energy demand hours is expected to generate large power fluctuations affecting the power grid. This effect is amplified by the fact that the power output of a wind turbine is related to the cube of the wind speed. Knowing when to expect these power fluctuations is of great interest for grid operators.

The passing of synoptic weather systems is a driver for wind power fluctuations on a timescale of multiple days (Kempton et al., 2010; Grams et al., 2017). For example depressions and associated fronts (Ahrens, 1994) are related to the four day peak in the spectrum defined by Van der Hoven (1957). Synoptic systems can be sufficiently resolved in relatively coarse resolution ($\mathcal{O}(10\,\text{km})$) hydrostatic weather simulations. On the other hand, turbulence manifests as small-scale chaotic motion in fluid

dynamics (Batchelor, 2000). It is also well known that turbulence generated by wind turbines affects turbines downstream (Calaf et al., 2010; Meyers and Meneveau, 2012; Stevens and Meneveau, 2017; Porté-Agel et al., 2020; Lanzilao and Meyers, 2022). The effects of turbulence are partly taken into account in Large Eddy Simulations (LES) or in Reynolds Averaged Navier-Stokes (RANS) simulations (Chung et al., 2002). Their high computational cost does however pose a constraint on domain size and the duration of these simulations.

  Variations in wind speed can also arise from mesoscale weather systems. With their length scales ranging up to a hundred kilometres and timescales spanning from ten minutes to a few hours, mesoscale weather systems occupy an intermediary position between turbulence and synoptic weather systems. They comprise thunderstorms and organised convection, but also sea breeze systems, low-level jets and gravity waves (Orlanski, 1975; Nunalee and Basu, 2014). These phenomena cannot be rep-
resented by coarse grid reanalyses such as ERA5 and require a convection-permitting model that can be run at kilometer-scale such as WRF (Peña et al., 2018; Larsén and Fischereit, 2021; Porchetta et al., 2021) or COSMO-CLM (Helsen et al., 2020; Thiery et al., 2015; Brisson et al., 2016; Van de Walle et al., 2020) to be resolved. Both Hahmann et al. (2015) and Wang et al. (2019) have evaluated kilometre scale climate models against in situ data and have shown the capacity of these models to reproduce mesoscale variability. Wind atlases utilising these models at a kilometre grid have been made available to the
public, such as the Dutch Offshore Wind Atlas (DOWA) (Wijnant et al., 2019) and the New European Wind Atlas (NEWA) (Petersen et al., 2014; Hahmann et al., 2020; Dörenkämper et al., 2020), and provide thoroughly validated information at the mesoscale level (Kalverla et al., 2020). Previous research has already shown that mesoscale weather systems are a driver for wind speed variability, both in an onshore as an offshore context. The offshore part of pure sea breezes can have an influence on the power output of a wind farm as it, in general, opposes or reinforces the synoptic wind flow (Steele et al., 2015). During
the night land breeze systems have the potential to generate offshore mesoscale wind speed variability (Gille et al., 2005; Short et al., 2019). Convective systems are also known to affect wind speed variability and a correlation between rainfall and wind speed variability is apparent (Weusthoff and Hauf, 2008; Trombe et al., 2014). A class of convective systems of which the conditions for formation, evolution and stability are not completely understood yet are the mesoscale convective systems (MCS) (Houze Jr, 2004). An overview of the different theories explaining the mechanisms behind MCS is summarised in for instance
the introduction of Short et al. (2023). The land-sea transition can lock these MCSs in place (Xu et al., 2012), potentially affecting offshore wind farms. In a coastal environment, variability is in general found to be dependant on the flow direction: when in autumn and winter the wind is coming from over sea rather than over land a larger variability is found over the North Sea (Vincent et al., 2011). The relatively warm sea water combined with cold air aloft creates unstable conditions, which result in mesoscale systems creating wind speed variations. But also in stable conditions systems can induce wind speed variability,
for instance due to nocturnal jets or gravity waves (Allaerts and Meyers, 2018).

  In our paper we use a 10-year integration of the convection permitting climate model COSMO-CLM with a horizontal resolution of 1.5 km to further study the atmospheric mechanics behind mesoscale variability. Augmenting existing research, we added here on the implications for power production for a full 10-year period. Moreover, this paper introduces a new index

which identifies spatially coherent mesoscale systems, allowing for detection of situations with a strong mesoscale system. We add on this spatial index with temporal analysis methods, since these methods are inherently complementarity. Additionally, this paper uses COSMO-CLM, which complements the frequently used model WRF, enabling a multi-model approach to applications in the wind energy sector in the future. Our analysis has been performed using 10-minute wind speed data, which is not available from wind atlases in our region of interest. The goals mentioned above can only be achieved if the model represents the real atmospheric winds. Therefore, we have incorporated an evaluation of the near-surface winds using Advanced Scatterometer (ASCAT) data (KNMI, 2018) and the 100-m winds using lidar data. These nicely complement each other as the ASCAT measures multiple locations, but only at 9h UTC and 21h UTC, while the lidar data provides measurements during the whole day, albeit at one single location. In order to minimise cable and maintenance costs offshore wind turbines are often built in farms near the coastline (Milborrow, 2020), implying that they are subject to coastal weather effects. This is why the Kattegat area (fig. 1) is our region of interest as it features a very irregularly shaped coastline, making it particularly interesting for studying mesoscale weather systems and their impact on wind farm power production. Moreover, it is largely surrounded by land, giving ample opportunity to study how mesoscale weather systems developing over land influence offshore winds. In fact, an offshore wind farm is present in the Kattegat: the Anholt wind farm, exploited by Ørsted. The aim of this paper is to investigate what factors influence mesoscale wind speed variability, on what timescales this variability occurs, and how it affects wind power output in offshore wind farms.

The remainder of this paper is structured as follows: section 2 contains the specifications of the simulation domain and the model set up (2.1) and validation (2.2), as well as the explanations of the methods used for studying the wind speed and wind power variability (2.3 and 2.4). Next, the results are reported in section 3, where a comparison of the model output with scatterometer and lidar data (3.1) is followed by the results for both a temporal and a spatial analysis (3.2 and 3.3). The added value of a convection permitting simulation over ERA5 for wind and power resources is assessed. The main findings of this paper are summarised in the concluding section.

## 2    Methods

### 2.1    Model setup

The model we use is the COnsortium for Small-scale MOdelling-CLimate Mode (COSMO-CLM) non-hydrostatic limited-area atmospheric model (Rockel et al., 2008). COSMO-CLM is a community model which is continuously maintained and developed by its users, under the coordination of the German Weather Service (DWD). The dynamical core of this model solves the primitive thermo-hydrodynamical equations describing a compressible flow in a moist atmosphere (Doms and Baldauf, 2018) with a timestep of 10 seconds. The COSMO-model uses an Arakawa C-grid and a staggered Lorenz vertical grid with terrain following Gal-Chen coordinates. The horizontal grid is mapped out in rotated coordinates with a spacing of $0.0135°$. This corresponds to a horizontal distance of approximately $1.5$ km which allows for explicit representation of deep convection in the model. For shallow convection the dynamical core of the model is expanded with a shallow convection parametrisation of

Tiedtke (1989). Subgrid-scale turbulence is parametrised by a one dimensional diagnostic level 2.5 closure scheme based on a prognostic TKE equation (Raschendorfer, 2001; Schulz, 2008*a*, *b*). Further parametrisations are present in the model that take subgrid-scale processes regarding micro-physical cloud processes and radiative transfer (Ritter and Geleyn, 1992) into account. COSMO-CLM has proven to be an adequate tool for long-term convection-permitting simulations, allowing for a statistical analysis of mesoscale weather systems (Brisson et al., 2016; Thiery et al., 2015; Van de Walle et al., 2020). It has also shown its value for studying wind speed metrics (Nolan et al., 2014; Wiese et al., 2019; Akhtar et al., 2021; Petrik et al., 2021). The model is directly driven by the 31 km resolution ERA5 reanalysis data (Hersbach et al., 2018). The sea surface temperature (SST) is also provided by ERA5 and updated on an hourly basis. The diurnal cycle of the SSTs is at 1 to 2K relatively limited, and this is comparable to the diurnal cycle found in potential sea surface temperature of the Baltic Sea Physics Analysis and forecast provided by CMEMS (Lindenthal et al., 2023). More information about the nesting strategy can be found in appendix A. Some deficiencies in ERA5, like meridional variability of surface winds, and moist convection (Belmonte Rivas and Stoffelen, 2019) are better represented at the kilometre scale resolution. For practical reasons the calculation of the 10-year simulation has been divided into smaller periods, but using the restart files generated by COSMO these periods were initialised with a warm start. To account for the relaxation and spin up of the forcing data 20 grid points from every side of the domain are excluded from the analysis domain. The setup for this simulation has been used before to investigate the effect of wind farms on the regional climate in the German Bight (Chatterjee and van Lipzig, 2020).

## 2.2 Scatterometer and lidar validation data

The model was evaluated with L3 scatterometer data from the ASCAT instrument on the Metop-A satellite available from the Copernicus Marine Environment Monitoring Service (CMEMS) (KNMI, 2018). The Metop-A satellite has a Sun-synchronous orbit and scans the Kattegat at approximately 9h and 21h UTC. The ASCAT instrument from the Metop-A infers the 10-meter wind vector (J. de Kloe et al., 2017) over sea via the backscatter of the microwave radiation it emits in three different directions. Validation with ASCAT data has already been used for a variety of offshore wind datasets (Hasager et al., 2020; Duncan et al., 2019). Points flagged by CMEMS for poor quality are removed from the dataset. The scatterometer does not cover the Kattegat on every overpass it makes, yet a total of $299,503$ Wind Vector Cells (WVC), which is equivalent to $\approx 20,800,000$ aggregated (i.e. averaged) model grid cells, is available for comparison. As the scatterometer only works over water surfaces, it is not available in the vicinity of coastlines (Verhoef et al., 2012). In order to allow for a comparison, the COSMO-model output is aggregated to the 12km grid of the scatterometer data. A direct point-to-point comparison between ASCAT and model output, like RMSD, might underestimate the quality of the model to represent the mesoscale variability. In the absence of strong forcing over sea, it can easily happen for a model to reproduce a mesoscale system, albeit slightly shifted in time and/or space. The reproduced mesoscale system then results in two errors. First it induces an error over the place where it should have been but is not reproduced now, and secondly it induces an error over the place where it is now but should not have been, which is referred to as a double penalty (G. -J. Marseille and A. Stoffelen, 2017). That is why, apart from the root-mean-square deviation (RMSD), statistical methods to assess the distribution of wind speeds like the $25^{th}$, $50^{th}$ and $75^{th}$ percentile wind speeds are also used. The comparison of these distribution parameters has an added value compared to an evaluation of the mean, as

an erroneous distribution can still have a good representation of the mean due to error compensation. Evaluating distribution parameters is therefore more rigorous than only evaluating the mean.

Using scatterometer data only the near surface wind speeds at 9h and 21h UTC of our model can be evaluated, which is not necessarily representative for the hub-height wind speed. Complementary to the scatterometer, a validation by a light detection and ranging (lidar) device located 2 kilometres west of the Anholt wind farm is used. This is not an ideal location, since it might be affected by the wind farm which is not implemented in our model, but it is the only measurement at hub height in the Kattegat area. Therefore a sub period of the measurement campaign was used, one in which the Anholt wind farm was
still under construction and the normalised availability of operation wind farms increased from 12% to 36% (see fig. A3 for the availability of the Anholt wind farm). Note that eventually 111 wind turbines were operational in the Anholt wind farm. Moreover the wind mainly blows from the west in this area (Karagali et al., 2013), resulting in a lidar signal that is largely unwaked. The lidar provides 10 minute averages of the wind speed and direction between 2013 and 2014 on different height levels, and is offered by Ørsted. The 10 minute averages are compared with the instantaneous wind speeds of the COSMO grid
cell corresponding to the location of the lidar. Note that the COSMO wind speed values are a grid cell average. This spatial averaging, similar to the temporal averaging of the lidar data, should diminish the impact of wind gusts on the analysis. The model wind speeds are interpolated to the measurement height level of the lidar using the wind profile power law given by

$$V(h_{\text{lidar}}) = V(h_m) \cdot \left(\frac{h_{\text{lidar}}}{h_m}\right)^{\alpha}, \tag{1}$$

where the shear coefficient $\alpha$ is given by

$$\alpha = \frac{\ln\left(V(h_{m+1})/V(h_m)\right)}{\ln\left(h_{m+1}/h_m\right)}. \tag{2}$$

Here $h_{\text{lidar}}$, $h_m$ and $h_{m+1}$ are measurement height of the lidar, and the two COSMO output levels closest to the measurement height of the lidar. The agreement between the lidar data and the model output is quantified using the Perkins Skill Score (PSS) (Perkins et al., 2007). The PSS quantifies the overlap between two equally-binned probability distributions, and is calculated by taking the sum of the minimum of the two probability distributions over all the bins. Formally this is expressed as

$$\text{PSS} = \sum_{i=1}^{N_{\text{bins}}} \min\left(\text{lidar}_i, \text{model}_i\right), \tag{3}$$

with $\text{lidar}_i$ and $\text{model}_i$ being the bin values of respectively the lidar and the model probability distributions.

### 2.3   Metrics for temporal variations in wind speed

The spectral density of a signal is estimated using a periodogram, also referenced to as a spectrum, calculated with the Welch method (Welch, 1967). Due to the uncertainty in a signal such as a time series of the wind speed, we can only make an estima-
tion of the underlying spectrum. In the Welch method the 10-year time series of 10-minute interval wind speeds is cut using a Hann window in overlapping sections of approximately seven days (1024 output intervals) in our case, with an overlap of

50%. This window length allows for part of the synoptic peak in periodogram to be seen. Subsequently the spectral density of every section is calculated using a Fast Fourier Transform (FFT) algorithm. The Hann window reduces the reflections arising from performing an FFT on a finite time series (Blackman and Tukey, 1958). The final mean spectrum is then calculated as the average of the spectrum of every section. This last step averages out the fast natural variability of the spectrum and results in an estimate for the true 10-year mean spectral density of the signal.

The periodogram is calculated for the four meteorological seasons allowing for a comparison between different seasons. The typical periodogram for a certain season (for instance spring, comprising the months March, April and May) is then obtained by taking the mean of the periodogram of that season over every year of the simulation (in this example taking the mean over all the spring periods in the years 2010-2019). For the winters of 2010 and 2019 the months December and the months January & February are respectively not included in the spectrum. Calculating the combined periodogram of these disjunct time periods is useful to depict the seasonal effect in wind variability. A Student's-t test on the 95% confidence level is used to quantify if the differences between the winter and the summer spectra are significant over a given period.

With the method described above an average periodogram of the 100 metre wind speed is calculated for every grid point for each season. These averaged periodograms can be integrated over a time interval of interest resulting in one single value per grid point that quantifies the temporal variability over that time interval. This makes it possible to visually compare the different grid points for each season and allows us to focus on specific scales, which would be challenging in the time-domain.

Periodograms can also be used to examine the fluctuations in potential wind power. The power curve of a wind turbine converts a wind speed time series to a wind power time series, and of the latter time series a periodogram can be calculated. The power curve of the Siemens SWT-3.6-120 wind turbine is available in table form (Bauer and Matysik, 2022), and a cubic spline interpolation is used to obtain the power for every possible wind speed. As the hub height for this type of turbine is 90 metres the 80 metre and 100 metre model wind speeds are interpolated to 90 metre via the power law given in equation 1.

## 2.4 Metrics for spatial variations in wind speed

We introduce the Mesoscale Spatial Variability Index (MSVI), extracted from the horizontal wind field at 100 meter, by sliding two square windows of different sizes grid point by grid point over the study area (fig. 2). Within both of these windows the mean wind speed is calculated. The small window aims to estimate the mesoscale wind speed, while the large window follows more or less the synoptic background. We define the MSVI as the deviation of the ratio between these two wind speeds with one:

$$\text{MSVI} = \frac{\langle v \rangle_{\text{small window}}}{\langle v \rangle_{\text{large window}}} - 1 \tag{4}$$

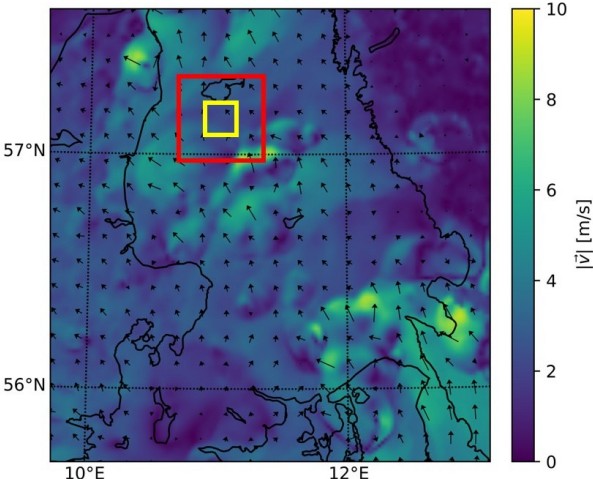

**Figure 2.** Wind speed and direction at 100 metres. The squares illustrate the windows defined for the Mesoscale Spatial Variability Index (MSVI) calculation. For this (and every other) hourly time step, both windows slide over the whole simulation domain and the maximum MSVI is saved.

and returns a dimensionless number quantifying how much larger mesoscale wind speeds are relative to the synoptic background.

One limitation is that this metric is derived using a model run of 1.5 km grid spacing, which is a slightly higher resolution than most wind atlases (e.g. DOWA Wijnant et al. (2019) and NEWA Petersen et al. (2014); Hahmann et al. (2020); Dörenkämper et al. (2020)), so the smallest signals that can be captured are at the effective resolution of $10 \times 10$ grid points (Kapper et al., 2010) ($\sim 15$ km). As size for the small window this effective resolution is chosen, thereby capturing most

210 of the mesoscale variability, although the smallest mesoscale signals are averaged out by this or any other product based on state-of-the-art mesoscale model simulations. The synoptic scale starts at $\sim 100$ kilometres (Oblack, 2020) but when taking this size for the large window, frontal systems were averaged out. Therefore, we took approximately half of this as dimensions for the large window to fully capture the synoptic background velocity field including the frontal systems. A size of 30 grid points ($\sim 45$ km) is chosen for the large window size, which is in area nine times larger than the small window. Using the large

window thunderstorms and smaller mesoscale systems are averaged out, whereas cloud clusters and fronts are not. With these window sizes, the MSVI turned out to identify mesoscale systems like convective systems and land-sea breezes (see section 3.3). This is also the scale of variability that can result in sudden power ramping events for wind farms. Even though this is not strictly speaking the mesoscale length scale (100 km), we will refer to it throughout the rest of the manuscript. As the small window is contained within the large window, an upper bound exists for the metric. Indeed: when the small window has

a mean wind speed of $x$ m/s, and the large window has a wind speed of 0 m/s everywhere else, but in the small window, the

denominator of the MSVI will be $\frac{1}{9} \cdot x$ m/s, giving an upper bound of eight to the MSVI.

Since the focus of this paper is on the offshore wind conditions, the MSVI is calculated only if at least 75% of the large window is sea, while all the onshore wind speed values are set to NaN. The maximum value per time step is calculated, quantifying the intensity of a mesoscale weather system in the domain during that specific time step.

A comparison between ERA5 and our simulation output was performed using the MSVI, assessing the added value of convection permitting simulations for mesoscale variability. This is done by regridding ERA5 to the grid of our COSMO simulation output, and then applying the MSVI metric to the ERA5 data.

Given the offshore wind speed variability, wind power fluctuations are expected. Wind speed and wind power are related to each other via the power curve of a wind turbine, and given this relation an MSVI analysis on the wind power could in principle be made. Yet the shape of this power curve imposes restrictions on our methodology: below the cut-in wind speed of a turbine (3 m/s) no power is produced, prohibiting a MSVI calculation for power fluctuations as the denominator would become zero. Instead we opt for a RMSD comparison between power time series for a stationary $10 \times 10$ window and a $30 \times 30$ window. The stationary windows are positioned in the area of the Anholt wind farm. The small window is in area nine times smaller than the large window. In order to cover the whole large window, nine small window power time series are calculated and compared to the large window power time series. The average over these nine RMSD values is then taken to assess the differences in wind power.

The Welch method and the MSVI metric complement each other. The Welch method produces a spectrum with information about wind speed variations over different timescales for every pixel. Bundling this information in a spectrum does however remove the temporal resolution of that time series. The MSVI on the other hand aggregates spatial information into a metric, and in doing so gives up spatial resolution. The temporal resolution in this method stays intact. As mesoscale systems are by definition bound in both space and time these two methods together offer a more complete view on offshore mesoscale wind speed variability than one metric would yield on its own.

## 3 Results

### 3.1 Evaluation with scatterometer and lidar data

The simulation is evaluated using scatterometer data. Over the 10-year integration we find a spatiotemporal RMSD between the 10 m wind speeds of 1.35 m/s, over the data points located away from coastlines and small islands. Note that the ASCAT wind speed error is about 0.5 m/s (Vogelzang and Stoffelen, 2021) with no particular expected regional deviation in the COSMO domain (Belmonte Rivas and Stoffelen, 2019). Moreover, the 1.35 m/s RMSD may be explained by the double penalty mentioned earlier, but there is no straightforward way of testing this. However, tests with spectral nudging did not substantially

improve the performance indicating that the lateral boundaries to a large extent control the timing and location of weather systems for this domain and model configuration. Distributions of wind speeds are compared via their 25[th], the 50[th] and the 75[th] percentiles (fig. 3). For the 25[th] and to a lesser extend the 50[th] percentiles, simulation and observation disagree, especially near the coastlines and over the islands. On the western shores simulation output and scatterometer agree better than on the eastern shore. The simulation also shows lower wind speeds over the Anholt and the Læsø islands. Recently, ASCAT winds have been validated extensively close to the coast and an operational product has been introduced providing good quality winds as close as 10 km from the coast (Vogelzang and Stoffelen, 2022), which could in the future be used to further evaluate the discrepancies between simulation output and scatterometer data. Relative differences between the simulation and the scatterometer further away from the islands and the coastline are smaller: -13% for the 25[th] percentile, +1% for the 50[th] percentile and +5% for the 75[th] percentile.

For validation of the model output above the 10 metre level a lidar device measuring the wind speeds at 102.6 metre is used. The 10 minute model wind speeds at the 80 m and 100 m level are extrapolated to 102.6 metre using the wind profile power law (equation 1) and then compared with the lidar data. The nearby Anholt wind farm was being built during the lidar measurement campaign and wasn't fully operational (wind farm availability is shown in figure A3). In order to minimise the impact of the Anholt wind farm on the validation of the model, we have also made a comparison taking only into account the first 100 days of the measurement campaign. Over the whole 2-year-long lidar data set the normalised PSS of the simulation output compared with the lidar data is 0.97 (fig. 4). During this period the simulation output overestimates the lidar data by on average 0.08 m/s. Taking only the first 100 days into account results in a similar normalised PSS of 0.96, with COSMO overestimating the lidar wind speeds by slightly less, namely 0.02 m/s. Even though there is some uncertainty about the effect of the wind farm on the lidar data, the small difference in performance between the two periods, together with the close correspondence between lidar data and COSMO gives us some confidence that the model performance is adequate. It might however be possible that COSMO under predicts winds in the simulation without turbines, which is masked by the wind farm wakes experienced by the lidar. However, the effect is likely small, due to the wind mainly blowing from the west in this area (Karagali et al., 2013).

## 3.2 Temporal variations

Using the Welch method the spectrum can be estimated for the 10-year 100 meter wind speed time series of each pixel in our domain. Differences between winter and summer are investigated using the combined periodogram for all winter months December, January and February (DJF) and for all summer months June, July and August (JJA ; fig. 5). The resulting periodograms resemble what is found in literature by Larsén et al. (2016) for 100 meter offshore wind speed spectra. For the lower frequencies the periodogram seems to be leveling off, which is indicative of the synoptic weather peak (Larsén et al., 2016). Comparing the simulation output with LiDAR data it appears however that our simulation slightly underestimates the intensity at the higher frequencies (figs. A1 and A2). These periodograms are averaged over a subdomain of the Kattegat, excluding coastlines and islands. On the high frequency end of the periodograms, a higher intensity is found in DJF compared to JJA. Relatively warm SSTs in winter result in turbulence and unstable conditions. Unstable conditions are a driver for convective

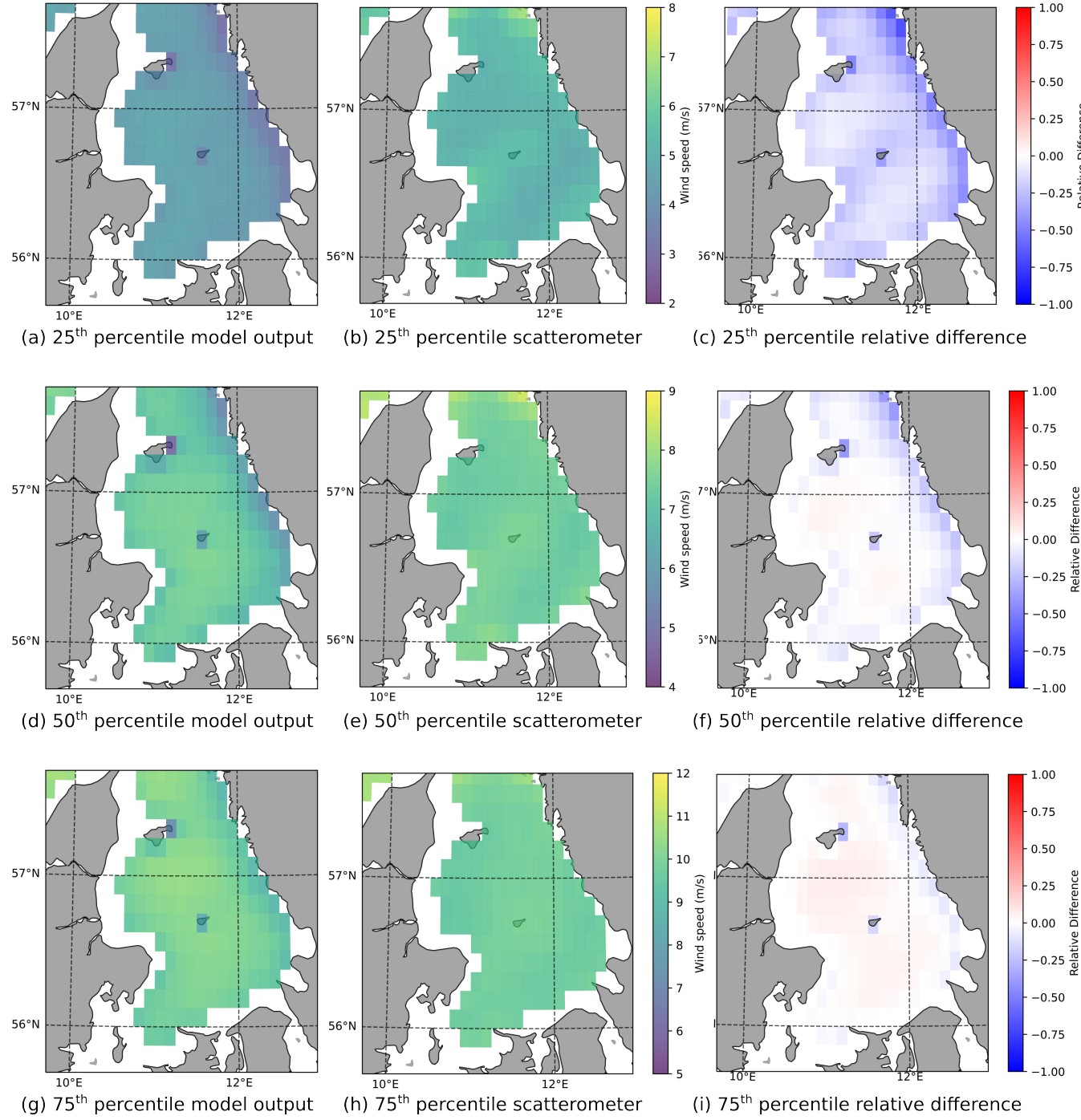

**Figure 3.** Comparison between the simulation and the scatterometer data. In the right most column the relative differences between the model output and the scatterometer data are plotted (model output - scatterometer data divided by the model). Note the difference in scales between different percentiles.

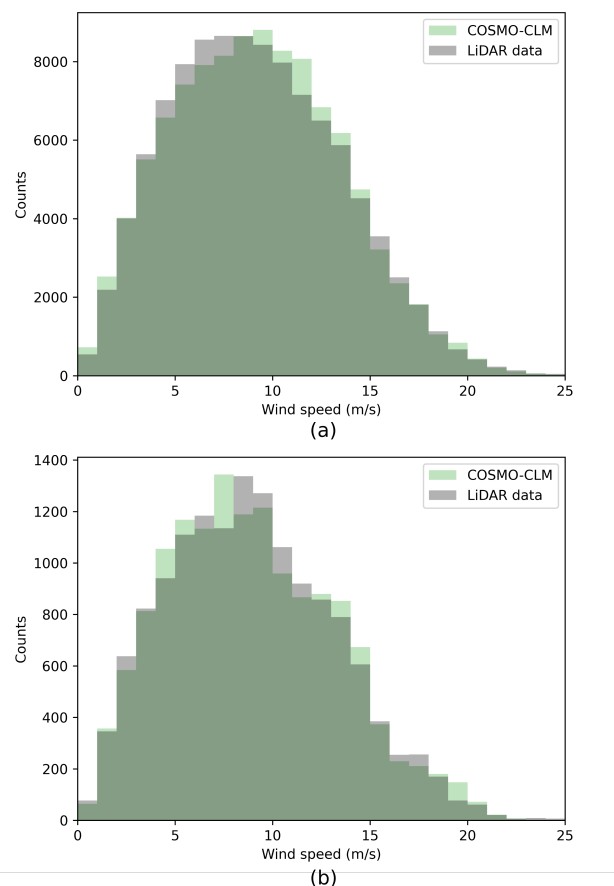

**Figure 4.** Distributions of both the model and the lidar wind speeds at 10 minute intervals at the position indicated in fig. 1. In **(a)** the whole 2 year measurement period is taken into account, in **(b)** only the first 100 measurement days are used when the wind farm was less than 50% operational.

cells. The systems are not bound to a specific place and are advected with the wind. Therefore they can move relatively fast and result in short timescale variability (20min-1h) (Ahrens, 1994). For wind speed variability on the long timescales (6h-12h), however, there is a larger intensity found for JJA. Using a Student's t-test we find that the differences over these time slots are significant at the 0.05 level. The difference between DJF and JJA on the longer timescales may be due to the sun being higher in summer, and it heating the land more effectively. With the sun over land, the air above it expands and generates a breeze over the sea during the morning or afternoon. During the night, due to the land cooling nocturnal jets may be formed. Sea breezes are fed by the contrast in land surface temperature and sea surface temperature. This keeps them confined to the vicinity of the intersection between land and sea, and this results in longer timescale variability (Ahrens, 1994).

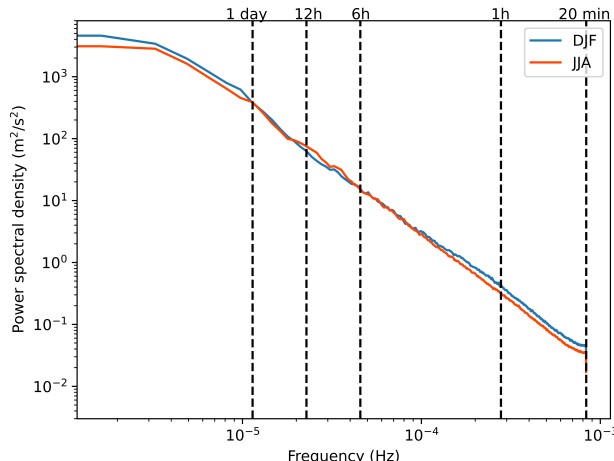

**Figure 5.** Periodogram of the 100 meter wind speed estimated using the Welch method for both DJF and JJA.

Next, periodograms for each grid cell are integrated over a chosen time slot, resulting in a map that quantifies the wind speed variability, allowing for a spatial analysis of the temporal wind speed variability. In the previous paragraph, a stronger variability was already found in JJA than in DJF for the long timescale winds (fig. 6). In contrast, for the short timescale winds, there is a stronger variability in DJF than in JJA (fig. 6), albeit an order of magnitude weaker than the long timescale variability. During winter the variability increases from southwest to northeast. This gradient is probably related to southwest prevailing winds and indicates that the convective systems need some spin-up time and distance to reach their maximal wind speed variability. Indeed, when isolating a time series of 11 consecutive winter days with only easterly winds (directions from 45° to 135° ), a clear gradient is observed from east to west (see fig. C1), confirming that spatial spin-up of convective systems is an underlying cause for the large wind speed variability over the northeastern part of the Kattegat. Similarly Vincent et al. (2011) relates wind speed variability over the North Sea in winter on these short timescales to unstable conditions created by the relatively warm sea water and the relatively cold air above. Their higher variability found towards the centre of the North Sea, compared to the coastlines, also indicates the spin-up time needed for the atmospheric conditions to reach maximal variability.

### 3.2.1 Integrated periodograms for the potential wind power

The combined periodogram for potential wind power integrated for a time slot running from 6 hours to 12 hours depicts a similar picture as for the wind speed (fig. 6 and fig. 7). Also for potential wind power, the variability is larger in JJA than in DJF. Potential wind power variability on shorter timescales from 20 minutes to one hour, however, differs from wind speed variability: stronger variability is found in summer (compare figs. 6 and 7). The discrepancy between wind speed variability and potential wind power variability is due to the particular shape of the power curve. This can be explained using the yearly mean wind speed and the power curve of the wind turbine used in this example (fig. 8). In winter, wind speeds are comparable

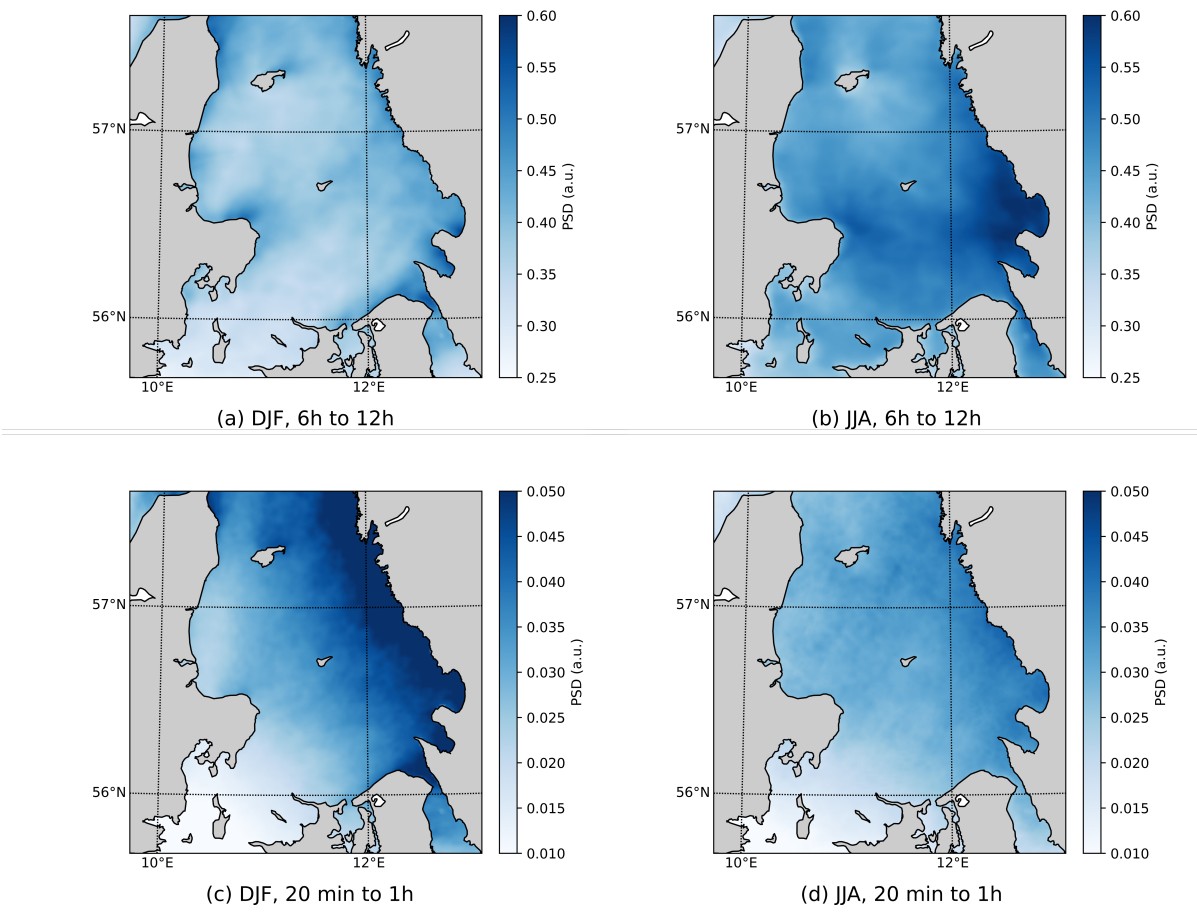

**Figure 6.** Integrated periodograms for both winter (DJF) as for summer (JJA). **(a, b)** On the long timescales there is a higher intensity in JJA than in DJF. **(c, d)** On the short timescales a higher intensity is found in DJF. This elevated intensity is probably related to convection triggered by the unstable conditions in DJF.

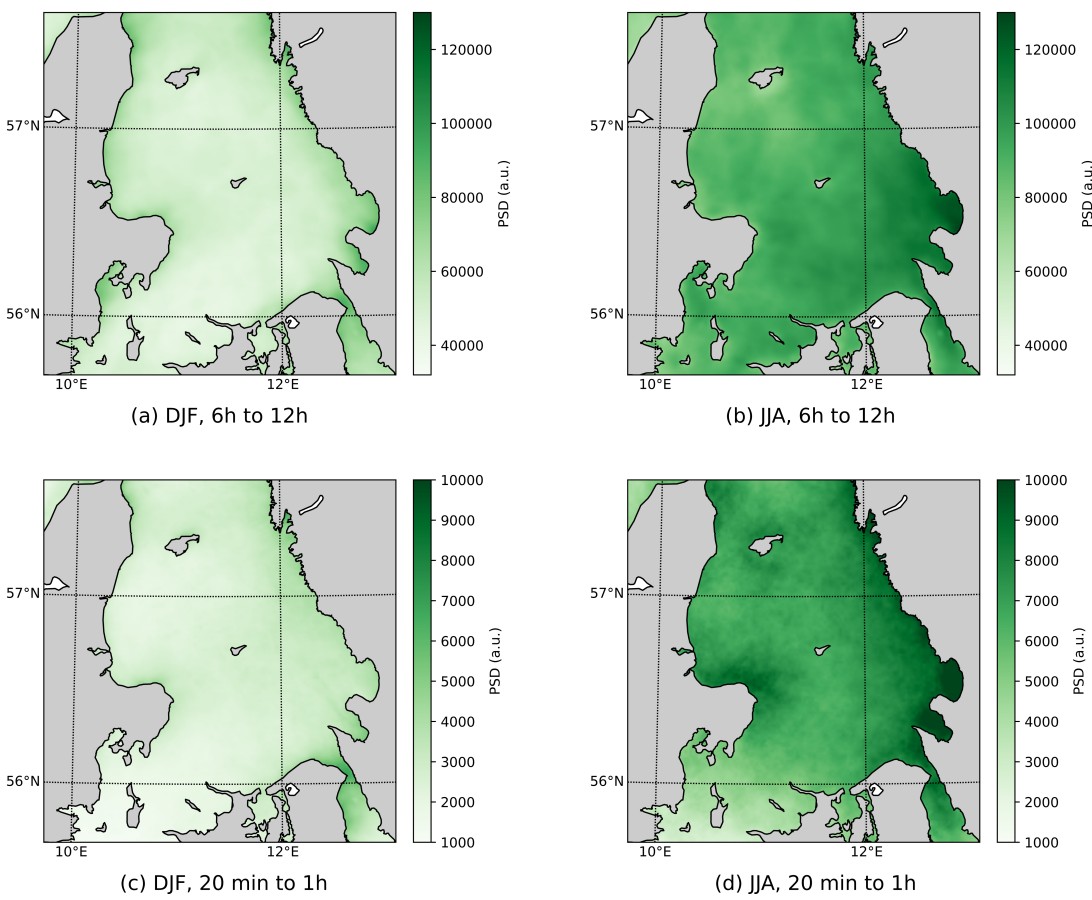

**Figure 7.** Integrated periodograms of wind power time series for both winter (DJF) as for summer (JJA). **(a, b)** As in figure 6 (a, b) the highest intensity on the long timescales is found in JJA. **(c, d)** Contrary to figure 6 (c, d) the larger variations in potential wind power are found here in JJA, while the larger fluctuations in wind speed on these timescales are found in DJF.

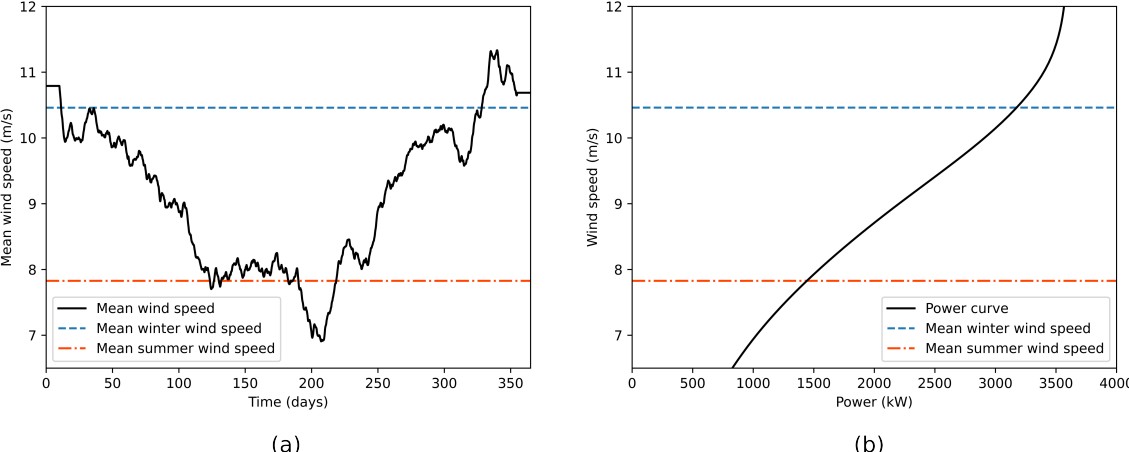

(a)                                    (b)

**Figure 8.** (**a**) Yearly mean 90 m wind speed over the Kattegat and (**b**) the power curve of the Anholt wind farm turbines. The power curve of a wind turbine relates a wind speed to a power output for that wind turbine.

to the rated wind speed of the turbine. Between the rated wind speed and the cut-out wind speed the power output of the turbine stays constant, so fluctuations around this speed do not result in fluctuations in the potential wind power. In summer, the mean wind speed is lower and closer to the regime where the potential wind power is proportional to the cube of the wind
speed, resulting in large fluctuations in potential wind power. On the shorter timescales fluctuations in wind speeds thus do not necessarily translate to fluctuations in potential wind power.

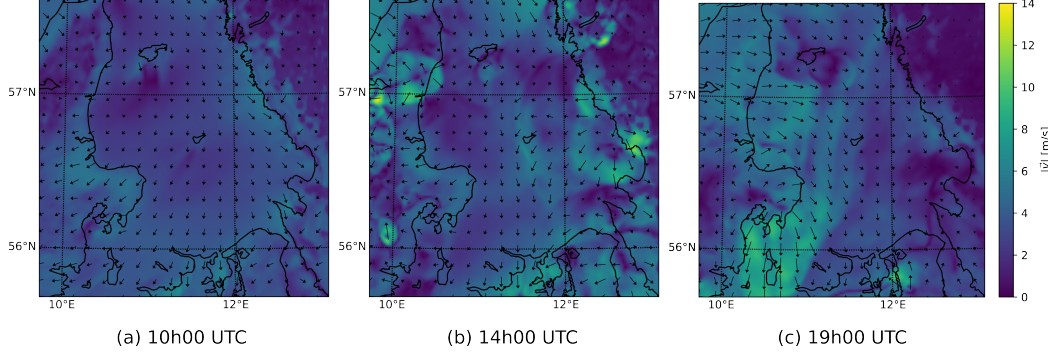

(a) 10h00 UTC              (b) 14h00 UTC              (c) 19h00 UTC

**Figure 9.** 100m wind speed of a convective system travelling over the Kattegat on June 25, 2014. The system is accompanied by local showers. A movie of this system is provided in Supplementary material 1.

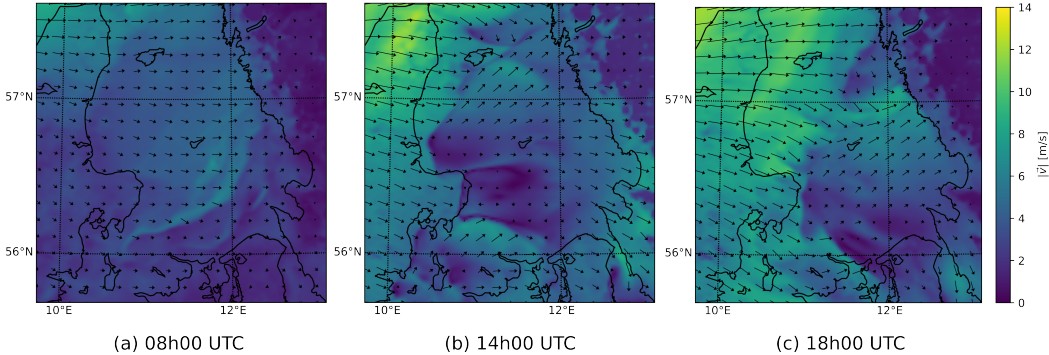

**Figure 10.** 100m wind speed of a sea breeze system over the Kattegat on August 28, 2013. While there is no onshore front, the system still affects the wind speeds over the Kattegat. A movie of this system is provided in Supplementary material 2.

## 3.3 Spatial variations in wind speed

The MSVI metric quantifies spatial variations in wind speed, and high MVSI values should indicate the presence of a mesoscale weather system. Indeed, looking at the wind fields associated with these peaks, a variety of mesoscale systems are clear, such

as convective and sea breeze systems developing over the Kattegat. Convective cells are initiated over land and the Læsø island in the afternoon, resulting in local showers and offshore wind speed variability (fig. 9) . Other peaks relate to sea breeze systems also causing spatial wind speed variability (fig. 10). In the case study, no onshore front is formed due to strong synoptic winds. However, this system still has a large impact on offshore winds, with a substantial drop in wind speed where the sea breeze counteracts the synoptic flow. At sunset the sea breeze systems detach and the synoptic wind over the Kattegat recovers.

Convective and the sea breeze systems are quite different, yet they both create mesoscale wind speed variations quantified by the MSVI.

Both the mean MSVI and its diurnal amplitude is higher in summer (JJA) than winter (DJF) (fig. 11). In winter, wind speed variability is found to be more or less constant throughout the day. In summer the MSVI clearly peaks in the afternoon, when

the surface temperature over land reaches its maximum (Jensen, 1960), confirming an influence of the land on the mesoscale winds over sea. Over at least one area over sea the mesoscale wind speed is on average 20% larger than the synoptic wind speed at 17h UTC in summer. Moreover, when analysing individual cases in our dataset with a high mesoscale wind speed component, it can be seen that the systems detected in summer often originate over land and then travel over sea. Models that explicitly resolve convection feature more mesoscale spatial variability in wind speed compared to coarser model data sets as

for example the ERA5 reanalysis. Calculating the mean MSVI for ERA5 over our simulation domain results in a substantially lower MSVI, and a smaller diurnal amplitude. To confirm that this is indeed the inability of a coarser scale gridded model to resolve mesoscale variability we have conservatively remapped the output of our convection permitting COSMO simulation to the ERA5 grid and then calculated the MSVI, which also resulted in a substantially smaller MSVI than the original simulation.

The difference between the coarser scale COSMO MSVI and the ERA5 MSVI on the same resolution might be due to the fact that the effective resolution of a model is substantially coarser than the native resolution. By aggregating to a coarser scale still part of the advantage of a high resolution model remains as earlier demonstrated by Brisson et al. (2016) for precipitation in convection permitting simulations.

The power output over the ten year simulation period is comparable for the small and the large window time series. The additional wind speed variations that the small windows capture cancel out over the simulation duration. The RMSD between the nine small windows on the one hand and the large window on the other hand is however quite substantial. At 286 kW this is 14% of the average power output. The wind speed variability captured by the mesoscale window translates to a non-trivial portion of the wind power variability. Therefore accurate forecasting of power fluctuations requires accurate high resolution wind forecasts.

## 4   Conclusions

We have established a convection-permitting simulation over the Kattegat area to investigate the mesoscale variability of off-shore conditions and its influence on wind power fluctuations. The simulation showed good agreement with scatterometer observations away from coasts and small islands with a spatiotemporal root mean square difference of 1.35 m/s, which is comparable to for instance Wang et al. (2019). As scatterometer products become more and more accurate near coastlines (Vo-gelzang and Stoffelen, 2022), future research will be able to study the nature of the discrepancies between simulation output and scatterometer data. Also at the 100-metre level, the 10-minute wind speed over the 2 years of available lidar data were well represented by the COSMO simulation. The bias is 0.08 m/s and the PSS, which is the overlap between the lidar and COSMO wind speed probability distributions, is 0.97.

The temporal variability was quantified over different timescales by integrating the Welch spectrum. Our results show that more variability in wind speed is expected in winter due to unstable conditions over sea. These unstable offshore conditions result in increased turbulence and induce convective systems, which generate wind speed variability on short timescales (20 minutes to 1 hour). The maximum variability is found in the northeastern part of the Kattegat, since southwesterly winds are prevailing. The variability on long timescales (6 hours to 12 hours) is found more pronounced in summer than in winter, prob-ably due to the development of sea breeze systems.

The power curve of an offshore wind turbine converts the wind speed data to power output data which, opposite to wind speed fluctuations, show that the energy production is more subject to fluctuations in summer than in winter. In summer the mean wind speed is situated in a regime where power output is very sensitive to wind speed fluctuation. In winter the mean wind speed is closer to the rated wind speed of a turbine, where the mean power output is less sensitive to wind speed fluctua-

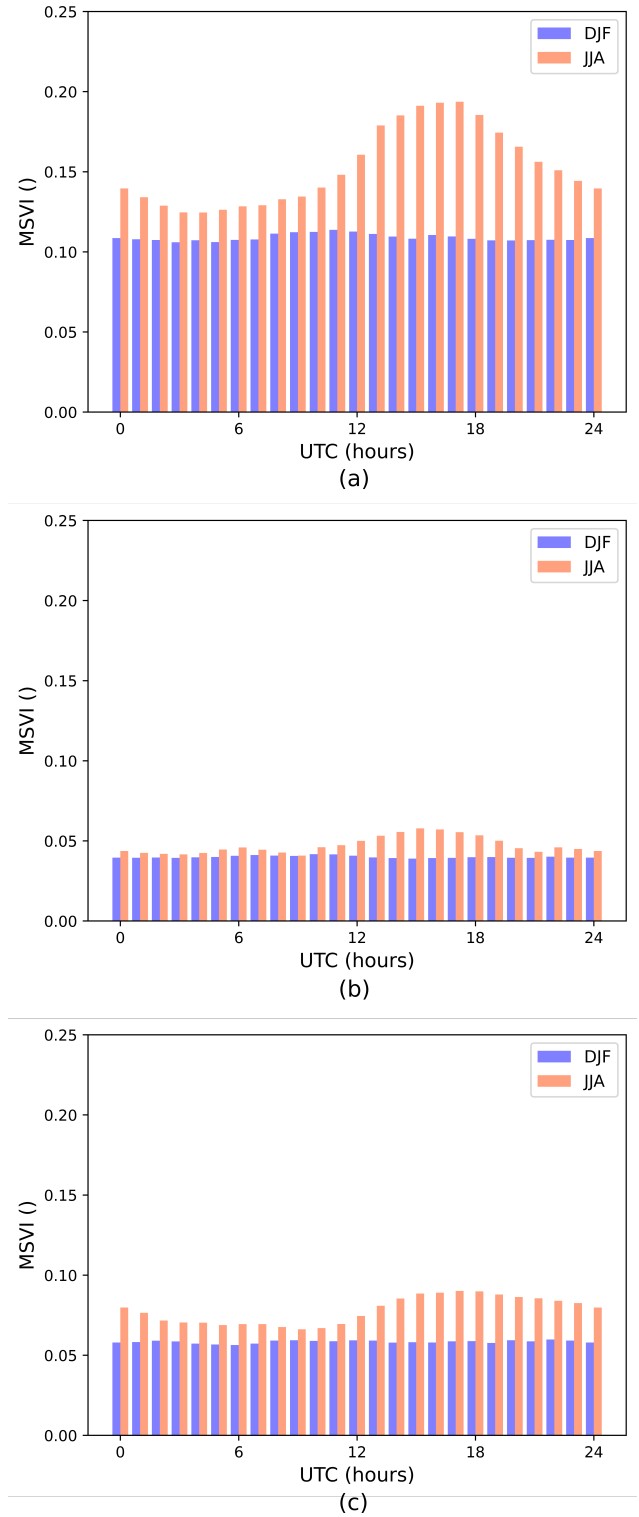

**Figure 11.** Hourly distribution of the MSVI for both winter (DJF) and summer (JJA). **(a)** MSVI of the high resolution COSMO simulation output. **(b)** MSVI of the ERA5 reanalysis data. **(c)** MSVI of the COSMO simulation output regridded to the ERA5 grid.

tions.

The offshore mesoscale spatial variability was studied using the MSVI defined in section 2. The MSVI is the deviation of ratio between the mean wind speed in a small window and a large window from one, calculated for every offshore position in the domain. The maximum MSVI in the Kattegat per time step is used to study the mesoscale wind speed variability. On average over a 10 year period the mesoscale wind speed in the Kattegat is up to 20% larger than the synoptic wind speed at 17h UTC during summer, exhibiting a clear diurnal cycle of systems forming over land transported over sea. Coarser grid models, as for instance ERA5, do not capture this variability and subsequently underestimate the MSVI. Spatial scales of modern large wind farms and mesoscale weather systems are similar, therefore the whole power grid becomes subject to power production fluctuations of a single farm. Though mesoscale climate modelling at the kilometre scale does not improve the mean power production estimation for the Kattegat, it is essential to correctly represent substantial power production fluctuations. The same argumentation is valid for resource assessment and climate change projections. Although mean power production might be well represented at the 30 km scale, yet models at the km scale are needed to gain insight in the variability at the hourly timescales.

The methods used in this paper can be applied to other regions. Research by Grams et al. (2017) has shown the importance of diversifying the deployment regions for wind energy in order to provide a stable electricity supply. Regions such as for example the Baltic Sea and the Mediterranean will become important to fill the gaps in the wind energy supply. Since most of the Mediterranean is not suited for fixed foundation wind turbines, this area has hardly been exploited for offshore wind energy. Further developments in floating wind turbines, which can operate at larger depths, could open up many opportunities for the Mediterranean, and other deep seas surrounded by complex coastlines.

*Code and data availability.* The code and data used to generate the figures 2-11 can be retrieved as a data set at dio.org/10.5281/zenodo. 10889808. The ERA5 reanalysis data used to drive the model simulation are available at the German Climate Computing Center (Deutsches Klimarechenzentrum, DKRZ). The ASCAT data was retrieved from the Copernicus Marine Service via doi.org/10.48670/moi-00183. The Anholt lidar data was used under an NDA, and is not publicly available.

*Author contributions.* Jérôme Neirynck contributed to the conceptualisation and methodology, the curating of data, the analysis of the data, the project administration, the visualisation and the writing of the paper. Jonas Van de Walle contributed to the methodology and simulation setup, the interpretation of the data, the visualisation and the writing of the paper. Ruben Borgers and Sebastiaan Jamaer contributed to the methodology, the interpretation of the data and the writing of the paper. Johan Meyers contributed to the conceptualisation and methodology, the data collection, the interpretation and discussion of the data, and the writing of the paper. Ad Stoffelen contributed to the methodology, the data collection, the interpretation and discussion of the data and the writing of the paper. Nicole van Lipzig contributed to the conceptu-

alisation of the study, the methodology, the data collection and interpretation, the overall supervision of the research project, and the writing of the paper.

*Competing interests.* One of the co-authors, Johan Meyers, is an associate editor of the WES journal. The authors have no other competing interests to declare.

*Acknowledgements.* The authors acknowledge support from the Research Foundation Flanders (FWO, grant no. G0B1518N), and from the project FREEWIND, funded by the Energy Transition Fund of the Belgian Federal Public Service for Economy, SMEs, and Energy (FOD Economie, K.M.O., Middenstand en Energie). The computational resources and services in this work were provided by the VSC (Flemish Supercomputer Center), funded by the Research Foundation Flanders (FWO) and the Flemish Government department EWI. Ad Stoffelen is supported by the EUMETSAT OSI SAF and the Copernicus Marine Service, providing the scatterometer winds. The authors would also like to thank the editor and the reviewers for their insightful comments and helpful suggestions which greatly improved the quality of the manuscript.

## Appendix A: Nesting strategy

Nesting strategies for regional climate modelling at the kilometre scale can vary greatly from model to model (Prein et al., 2015). Many studies use a 1:10 resolution jump, but a 1:20 resolution jump has been done before. In previous work that we did with COSMO-CLM, a clear benefit was found in the representation of mesoscale weather systems by directly nesting within a 25 km grid spacing host domain to 2.8 km, instead of using nesting steps in between (Brisson et al., 2015).

Different simulation setups have been tested for 3-month integrations. These tests included a larger domain ($340 \times 360$ grid points compared to $180 \times 184$ grid points), adding an intermediate nesting at $\approx 12$ km resolution and applying spectral nudging. We found no added value of using a larger domain, of adding an in between nesting step and of applying spectral nudging compared to the wind data from scatterometer. This result is in line with the findings of Ban et al. (2021) where different nesting strategies of COSMO-CLM in ERA-Interim do not show any substantial differences. For the temporal mesoscale variance metric as defined by Vincent and Hahmann (2015) we found that the larger domain appears to have a slightly higher variance, but the difference between both simulations is rather small.

For the assessment of the mesoscale variance in our simulation we analysed the spectra of the LiDAR and the simulation 100 m wind speeds (fig A1). The dashed lines represent the timescales used to define the mesoscale variance in Vincent and Hahmann (2015). Over these timescales simulation and results appear to agree quite well, even though near the higher frequencies the spectra start to diverge. For the first 100 measurement days (fig A2) the agreement seems to be better, but due to natural variability not being averaged out in this relatively small dataset uncertainties become larger.

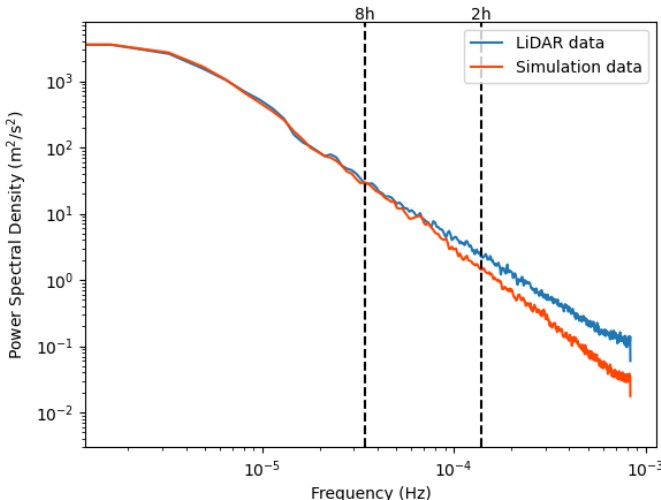

**Figure A1.** Spectra for the LiDAR and simulation 100 m wind speeds.

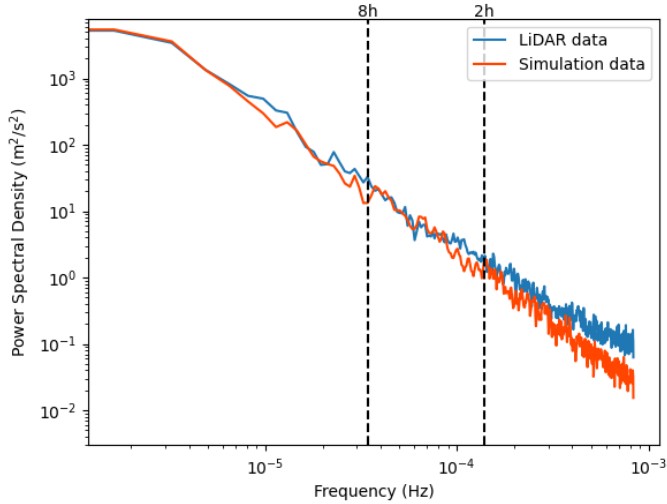

**Figure A2.** Spectra for the LiDAR and simulation 100 m wind speeds for the first 100 measurement days, when the Anholt wind farm was less than 50% operational.

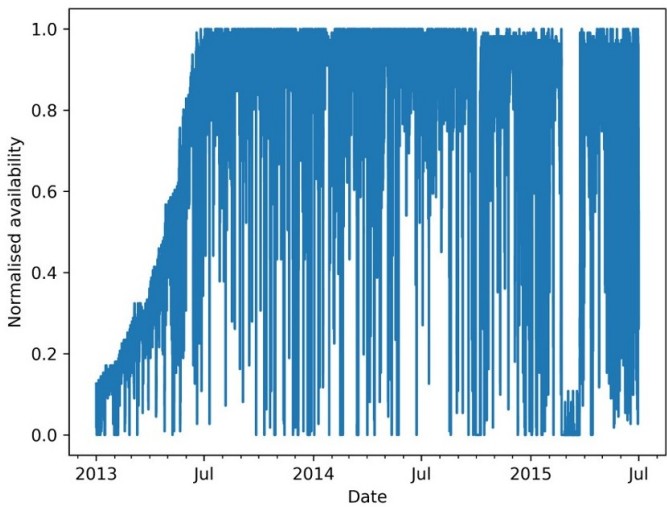

**Figure A3.** Normalised availability for the Anholt wind farm.

## Appendix B: Sea surface temperature

We used some test runs to get to an optimal simulation setup, and in one of these test runs the sea surface temperature (SST) was kept constant at the initial (January) level. When comparing the spectrum for both runs we find on longer timescales no difference the intensity. On the shorter timescales there is however a higher intensity for the simulation with changing SST. As the SST does not change significantly over the daily timescales this higher intensity on short timescales might be because of the average SST in winter being higher than the SST from the first of January, thereby inducing more convective situations. In summer we find a higher intensity for the longer timescales for the simulation with a constant January SST. An enhanced contrast between land and sea surface temperature might thus result in more wind speed variability on long timescales related to the land-sea breeze system.

## Appendix C: 11 day easterly winds

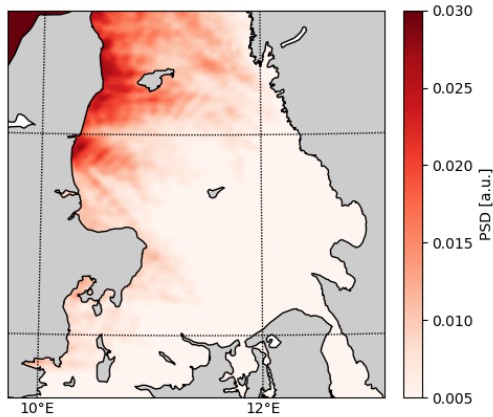

**Figure C1.** Integrated periodogram for an 11-day period with easterly winds over the Kattegat area.

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
