# Peer review of "Mesoscale weather systems and associated potential wind power variations in a mid-latitude sea strait (Kattegat)"

_Wind Energy Science, 2023_

## Author Comment (AC1)

**Reviewer 1:**

**General comments**

1. *I really like the attempt to separate the 'mesoscale' and 'synoptic-scale' winds. However, I'm not really sure what physical processes are being picked up by this approach. The 'mesoscale wind', is designed to align with the effective resolution of the model, rather than any physical arguments. The 'synoptic-scale wind', calculated over a 45kmX45km grid box, is still well within the stated length-scale of 100km for mesoscale weather systems. For example, a sea-breeze disturbance could cover the whole larger box, as could a large thunderstorm complex. Can the authors try to quantify/justify this choice of length-scales? Some kind of spectral analysis or spatial filtering with known spectral properties could be helpful here. Either the length-scales should be clearly justified by physical arguments about the scale of mesoscale weather systems, or by practical arguments about the scale of variability that is of relevance to large offshore wind farms.*

The size of the large window is chosen to be smaller than the length scale of the synoptic weather systems in the figure placed hereunder. It is true that the MSVI as we use it has some selectivity towards variability at a certain spatial scale, but this is also the scale of variability that can result in sudden power ramping events for wind farms.

[Figure]

2. *When discussing the variability that the authors call long time-scales (6-12 hours), the focus is on diurnal processes over the land influencing the wind over the sea. However, there can also be a diurnal cycle over the water, especially in shallow areas where the water can warm or cool more*

*rapidly. The authors did not state how the SSTs were specified in the model, nor whether they were being updated on daily or sub-daily time-scales.*

The sea surface temperatures used in COSMO are inferred from the ERA5 reanalysis dataset. The SSTs do change in our simulation on a sub-daily timescale, but the diurnal cycle of these SSTs is at 1 to 2 K relatively limited. This cycle is comparable to the diurnal cycle found in potential sea surface temperature of the Baltic Sea Physics Analysis and Forecast provided by CMEMS. As our simulation set up does not include an oceanic circulation model the complex structure of eddies found in the Baltic Sea Physics Analysis and Forecast is not present in the COSMO simulation.

In the model setup section we have added a statement regarding the SST's being provided by ERA5, and we have added a section to the appendix discussing our findings from a test run we did with a constant SST.

**2.1 Model setup**

The model we use is the COnsortium for Small-scale MOdelling-CLimate Mode (COSMO-CLM) non-hydrostatic limited-area atmospheric model (Rockel et al., 2008). COSMO-CLM is a community model which is continuously maintained and developed by its users, under the coordination of the German Weather Service (DWD). The dynamical core of this model solves the primitive thermo-hydrodynamical equations describing a compressible flow in a moist atmosphere (Doms and Baldauf, 2018). The COSMO-model uses an Arakawa C-grid and a staggered Lorenz vertical grid with terrain following Gal-Chen coordinates. The horizontal grid is mapped out in rotated coordinates with a spacing of 0.0135∘. This corresponds to a horizontal distance of approximately 1.5 km which allows for explicit representation of deep convection in the model. For shallow convection the dynamical core of the model is expanded with a shallow convection parametrisation of Tiedtke (1989). Subgrid-scale turbulence is parametrised by a one-dimensional diagnostic level 2.5 closure scheme based on a prognostic TKE equation (Raschendorfer, 2001; Schulz, 2008a, b). Further parametrisations are present in the model that take subgrid-scale processes regarding micro-physical cloud processes and radiative transfer (Ritter and Geleyn, 1992) into account. COSMO-CLM has proven to be an adequate tool for long-term convection-permitting simulations, allowing for a statistical analysis of mesoscale weather systems (Brisson et al., 2016; Thiery et al., 2015; Van de Walle et al., 2020). It has also shown its value for studying wind speed metrics (Nolan et al., 2014; Wiese et al., 2019; Akhtar et al., 2021; Petrik et al., 2021). The model is directly driven by the 31 km resolution ERA5 reanalysis data (Hersbach et al., 2018). The sea surface temperature is also provided by ERA5 and updated on an hourly basis. More information about the nesting strategy can be found in appendix A. Some deficiencies in ERA5, like meridional variability of surface winds, and moist convection (Belmonte Rivas and Stoffelen, 2019) are better represented at the kilometre scale resolution. For practical reasons the calculation of the 10-year simulation has been divided into smaller periods, but using the restart files generated by COSMO these periods were initialised with a warm start. To account for the relaxation and spin up of the forcing data 20 grid points from every side of the domain are excluded from the analysis domain. The setup for this simulation has been used before to investigate the effect of wind farms on the regional climate in the German Bight (Chatterjee and van Lipzig, 2020).

(Appendix) We used some test runs to get to an optimal simulation setup, and in one of these test runs the sea surface temperature (sst) was kept constant at the initial (January) level. When comparing the spectrum for both runs we find on longer timescales no difference the intensity. On the shorter timescales there is however a higher intensity for the simulation with changing sst. As the sst does not change significantly over these timescales this higher intensity on short timescales might be because of the average sst in winter being higher than the sst from the first of January. In summer we find a higher intensity for the longer timescales for the simulation with a constant January sst. An enhanced contrast between land and sea surface temperature might thus result in more wind speed variability on long timescales.

3. *The authors looked at both temporal and spatial variability, but it would have been nice to see a greater attempt to relate these results. In particular, given that the temporal and spatial analysis should be capturing the same thing, why was it necessary to work with the spatial methods that place limitations on coastal areas? Why could the integrated periodograms over 'mesoscale' and 'synoptic-scale' periods have been compared, in a similar way to the MSVI? This would have given well-resolved maps of where the mesoscale variability was playing an important role?*

Mesoscale systems are by definition bound in both space and time, and we tried to examine mesoscale variability in both space and time. In the spatial analysis there was a limitation placed on coastal areas in order to limit the influence of the wind speed-up when transitioning from land to sea. The results of the MSVI metric become harder to interpret when one of the windows contains a disproportionate amount of land pixels. The following statement on the complementarity that the spatial and temporal analysis have regarding one another is added to the methods section

The Welch method and the MSVI metric complement each other. The Welch method produces a spectrum with information about wind speed variations over different timescales for every pixel. Bundling this information in a spectrum does however remove the temporal resolution of that time series. The MSVI on the other hand aggregates spatial information into a metric, and in doing so gives up some spatial resolution. The temporal resolution in this method stays intact. As mesoscale systems are by definition bound in both space and time these two methods together offer a more complete view on offshore mesoscale wind speed variability.

**Specific comments**

1. *Page 3, line 41: "The effects of turbulence can be taken into account in Large Eddy Simulations" -> I think it should say "partly taken into account", since LES models only capture the larger part of the turbulence.*

This is changed in the manuscript.

2. *Page 3, line 53: "the majority of research is based on the onshore extent of the systems" -> There are a few references that look at the offshore part of land-sea breeze circulations that might be missing here. For example, Short et al. (2019) and Gille et al. (2005). In this context, the authors should also mention the land-breeze, which may be more relevant for offshore winds.*

These two references are indeed very interesting and are added to the manuscript

3. *Page 2, line 38: There are more recent versions of the wind speed spectrum that you could refrence - eg. Kang et al (2016).*

The data used in Kang et al. (2016) is taken from a continental location. For our paper we therefore opted to include a spectrum from a more coastal climate, such as Van der Hoven (1957) and Larsén et al. (2016).

4. *Page 3, line 45: Change first sentence to "Less is known about the impact of mesoscale weather systems on wind variability, for example in organised convection"*

This whole paragraph has been restructured and does no longer include the sentence: "*Less is known about mesoscale weather systems, for example in organised convection.*"

5. *Page 3, line 55: Add reference to Trombe et al. (2014).*

This interesting paper is quite relevant and is added to the paper.

6. *Page 3, lines 45-60: I think this section is missing discussion of a major source of mesoscale variability, which is from organised thunderstorms or MCS.*

This is added to the manuscript.

7. *Page 4, line 100: What height is the ERA5 wind speed accuracy quoted for?*

As suggested by the other reviewer, this statement is removed from the paper.

8. *Page 5, line 109: 'aggregated' - clarify - is this interpolated, or averaged?*

We have clarified in the manuscript that we mean averaged here.

9. *Page 5, line 130-133: The authors state that the time series is cut in overlapping sections. Later, it says that a 'Hann window is used to cut the signal into sections'. Is this duplication?*

This paragraph is slightly restructured to remove the duplication

10. *Page 5, line 134: 'fast natural variability' - is it 'fast', or just removing the noise?*

We can't really call it noise, as noise is considered an alternation to the original signal due to measurement imperfections. Our simulation should in principle not introduce something that can be classified as noise.

11. *Page 6, line 135: 'good estimate' -> how do you know it's 'good'?*

Good was meant to indicate that we consider the number of periodograms that are averaged to be large enough to converge to the underlying spectrum of wind speed variations. As this is never proven in the paper this statement of it being good is confusing, and "good" is removed from the sentence

12. *Page 6, line 137: 'integrated over a 3-month interval' - what does that mean?*

This is indeed a confusing statement so we removed it.

13. *Page 6, line 144: 'period' -> can the authors choose a different word? This could mean 'a period of time' or 'periods from the spectrum'.*

We have used "time interval" where possible to alleviate confusion for the reader.

14. *Page 7, line 169: suggest changing to "As the small window is contained within the large window'*

This is changed in the manuscript.

15. *Page 7, line 170: 'everywhere' -> 'everywhere else'*

This is changed in the manuscript.

16. *Page 10, line 220: The comparison with lidar data at high frequencies is mentioned, but as far as I can see, this is not shown in the figures.*

A reference to the figures of the comparison of the model periodogram and the lidar data in the appendix is added to the manuscript.

17. *Page 10, line 225-227: Why would the diurnal effects only show up on the 12h time-scale, and not the 24-hour time-scale?*

Masouleh et al. (2019) show that quite a large portion of sea breeze events have a duration of less than 12 hours. That is the reason why there is not one single peak in the spectrum shown in figure 5. While we believe that the temperature difference between land and sea is a driver for wind speed variability at these time scales (since the variability on these timescales is larger in summer compared to winter) the systems that this temperature contrast induces are apparently not always following this diurnal cycle.

18. *Page 11, line 230: It would be useful to contrast/compare the results to Vincent (2011).*

Indeed, the results of Vincent et al. 2011 are quite similar and a comparison is added to the manuscript.

19. *Page 11, line 240: The authors mention the issues of spin-up around the edges of the domain. This is an interesting problem, but raises the question of whether the boundary removed from the edges was sufficient. Can we really trust the results, give these effects?*

The appendix on the nesting strategy includes a small discussion on the tests we performed with a larger simulation domain. A reference to this discussion has been added to the model setup section.

20. *Page 11, line 246: Can the authors show the periodogram for power, as well as the integrated maps? The power time-series presumably has some constant sections where the wind speed is greater than 15 m/s or less than 3 m/s, and possibly some sudden jumps due to the cut-out speed being reached. What impact do these shocks have on the periodogram?*

These shocks elevate the spectrum quite a lot compared to the wind spectrum, but as these shocks don't necessarily happen at specific time intervals there are no clear peaks in this spectrum. The shape of the spectrum remains similar to the wind spectrum.

21. *Page 14, figure 9: I don't find this graph very useful. What is it supposed to be showing? Could the authors show it as an average annual cycle, averaged over the 10-year period? Or overlay a smoothed version so that the curve is more obvious?*

This graph is supposed to make clear to the reader that the MSVI results in a 1-D time series.      This point is clarified in the text, and the figure is removed from the paper.

22. *Page 15, line 269: Why are the 'mesoscale winds' always more than the 'synoptic wind speed'? This is attributed to the land-sea breeze circulation and other mesoscale phenomena. In some cases, this might not be the case, since if the sea-breeze opposes the background flow, then it will weaken the wind overall. I agree that usually, mesoscale phenomena would lead to more windy conditions, but it should not be assumed that this is the case.*

It is true that the MSVI detects a locally elevated wind speed in each time step. Some mesoscale systems do indeed generate lower wind speeds, but due to the local nature of mesoscale weather systems a place with elevated wind speeds should be found in the vicinity of that mesoscale system. While our method is definitely not perfect the MSVI has the potential to capture this variability in wind speed created over these relatively small spatial scales.

    *23. Page 15, lines 270-272: This section lacks references*

References have been added to the manuscript

---

## Author Comment (AC2)

**Reviewer 2:**

**Major Concerns:**

**Framing of the paper**

*To put it succinctly, I'm uncertain if the authors are claiming that either (a) few mesoscale wind simulations studies have been published or (b) if they are very specifically claiming that the mesoscale wind simulation studies that have been published have not sufficiently analyzed "mesoscale wind speed variability".*

*If they are claiming (a), then I strongly disagree. The authors categorize the following as mesoscale phenomena: "thunderstorms, but also sea breeze systems, low-level jets and gravity waves". Many many papers have been published on how wind energy is affected by these, e.g. (Tomaszewski and Lundquist 2020 for thunderstorms, sea breezes in the North Sea by Steele et al. (2014), many papers for LLJs, many papers by Allaerts on gravity waves). Additionally, there has been extensive work on the mesoscale through large-scale wind resource assessments (like the New European Wind Atlas and the WIND Toolkit) as well as smaller but important studies (e.g., Hahmann et al. 2015). Relatedly, I believe that NEWA, the Dutch Offshore Wind Atlas (Kalverla et al. 2020), and possibly NORA3 (Cheynet et al 2022) all include Kattegat in their simulations, and these products should be mentioned somewhere if so.*

*If (b), the citations in L62-63 are a small set of mesoscale wind resource papers, and I don't believe these papers do anything unique regarding wind speed variability.*

We have restructured and rewritten large parts of the paragraph that used to start with "*Less is known about mesoscale weather systems, for example in organised convection.*" We have added some references here and also mention the wind atlases. In our paper we perform some analyses which add to the understanding of factors that are relevant for mesoscale wind speed variability. We hope that rewriting part of the introduction results in a better framing for the paper.

**Novelty**

*I normally don't comment on novelty, but many mesoscale simulations have been conducted specifically in the North Sea, with domains that include Kattegat. A non-exhaustive list includes Hahmann et al. (2015) and the New European Wind Atlas (see the two papers in GMD). I believe that the Dutch Offshore Wind Atlas (see papers by Kalverla) and possibly NORA3 (Cheynet et al. 2022) may also cover this region. I don't see any references to these papers. The authors should clearly state why their work is novel to help the readers out. I feel like I've seen many papers also do similar-but-different seasonal analysis before (e.g., Wang et al 2019 in California), and these should also at least be referenced.*

We have set up our own simulation of this area since this gives us more freedom regarding the generation of output data. This is mainly important for the temporal variability analysis, which uses 10-minute interval output data. We also believe that the spatial and temporal analysis methods used in our paper aid the understanding of factors driving mesoscale variability. In addition to this we also plan on using this simulation setup for further studies, for which control of the model setup is required. As this last point is not relevant for this study it is (in contrast to the other points) not added to the manuscript.

**Methodology**

*Simulation setup: What is the timestep of the simulation? How frequently is data saved out? Did you run a single, 10-year long simulation or did you chunk up the job into smaller periods? Did you use spectral nudging to encourage the long simulation to not drift too far away from the expected ERA5 values? I'm unfamiliar with COSMO, but in WRF, it is known that the PBL scheme choice is very important. If multiple options are allowed in COSMO, which PBL scheme is used here? While not required by this journal, I highly encourage the authors to upload an example configuration file somewhere so that others may more easily reproduce this study in the future. There is possibly one on Zenodo, but the link is private, so I cannot view its contents.*

The simulation timestep is 10 seconds. The output frequency differs from variable to variable, but for the wind speed on specific height levels (10m, 80m and 100m) it is set to 10 minutes. For practical reasons the calculation of the 10-year simulation has been divided into smaller periods, but using the restart files generated by COSMO these periods were initialised with a warm start. Subgrid-scale turbulence is parametrised by a one-dimensional diagnostic level 2.5 closure scheme based on a prognostic TKE equation (Raschendorfer, 2001; Schulz, 2008). This is the standard PBL scheme used in COSMO-CLM. A file containing the configuration of the model will be added to the Zenodo, and made available to the public.

This information is added to the model setup section

*Validation efforts: Researchers often validate modeled winds against measured winds in order to built trust, but the validation study here instead erodes my confidence. I also don't see how validation furthers the authors stated goals in the intro. I believe that all the lidar analysis should be struck entirely. Please write a sentence or two that explicitly ties the motivation behind the validation back to the goal of the paper.*

The validation with lidar data is included in the paper in order to validate the wind speeds at 100 metre, as we mainly use these wind speeds in the paper. The lidar data is also a nice addition to the ASCAT data, as ASCAT measures multiple locations, but only at 9h UTC and 21h UTC, while the lidar data provides measurements during the whole day, albeit at one single location.

*Lidar: The simulated winds do not include wakes, but the lidar is probably being waked. That waking can be significant (>1 m/s modifications). We also don't know how close the lidar is to the nearest turbine, and also the number of operational turbines changes throughout the comparison period, changing the waking strength. The authors attempt to minimize the effects of waking by looking at periods when wind farm availability stayed below 50%, but that is insufficient in my opinion.*

The lidar is positioned 2 kilometres west from the Anholt wind farm. The wind farm was under construction during the measurement campaign, and the availability of the wind farm is shown in the figure hereunder (this figure is added to the appendix). The first 100 days of this measurement campaign see a rise in wind farm availability, but this availability stays under 50%. As there is not a large difference in model performance compared to lidar data for the first 100 days, relative to the full dataset, while there is a large difference in wind farm availability means that the impact of the wind farm on the lidar data is not that large. This does not mean that the Anholt wind farm does not have an

impact on the surrounding atmospheric conditions, but that due to the wind mainly blowing from the west in this area (Karagali et al. 2013), resulting in a lidar that is largely unwaked.

This discussion has been added to the manuscript.

[Figure]

*ASCAT: I am less familiar with this instrument, but I suggest that the authors point out that others in wind energy have used this data source before (e.g., Hasager et al. 2020). I mention this because I know there is some controversy regarding validating WRF against SAR (a different instrument), but if others in the wind energy sector have looked at ASCAT previously, then there is a stronger case for its use here.*

A reference to the validation of the DOWA with ASCAT has been added to the paper, as well as a reference to Hasager et al. 2020. A statement about this has been added to the manuscript.

*ASCAT: I haven't seen others make maps of winds at different percentiles. Why did you do this instead of simply comparing mean wind speed maps? You give a justification on L110, but I don't quite follow. Maybe if you provide a summary of what the "double penalty" is, that would help clarify things.*

A direct comparison between ASCAT and model output might underestimate the quality of the model to represent the mesoscale variability. In the absence of strong forcing over sea, it can easily happen for a model to reproduce a mesoscale system, albeit slightly shifted in time and/or space. The reproduced mesoscale system then results in two errors. First it induces an error over the place where it should have been but is not reproduced now, and secondly it induces an error over the place where it is now but should not have been (yet) (G. -J. Marseille and A. Stoffelen, 2017). That is why, apart from the RMSD, statistical methods to assess the distribution of wind speeds, like the 25th, 50th and 75th percentile wind speeds, are also compared. This statement has been added to the 2.2 section explaining the methods used in the validation of the model with scatterometer data.

*ASCAT: L106: It's hard to judge if 229,503 WVC is a lot of data or a small amount of data. Could you give more context? Maybe an easy number to calculate is the number of aggregated WRF grid cells over this period.*

This corresponds to approximately 20,800,000 grid cells of our COSMO model, and this statement has been added to the manuscript.

*ASCAT: If a model validates well at 10 m, that doesn't necessarily mean that the model is accurate at hub-height (see the Bodini extrapolation papers if you're interested). Please add that caveat in somewhere, as many today would consider validating against near-surface winds to be of minimal utility (for what it's worth, I am not one of those people).*

We have added to the paper that the ASCAT device only measures the near surface wind speeds at 9h UTC and 21h UTC and that the lidar validation has been added to the paper to validate the model further at 100m.

**Analysis**

*Fig 5: I really like the spectral analysis. One of the goals of this paper is to "to investigate what factors influence mesoscale wind speed variability", and as such, I would like to see a stronger physical justification as to why the wintertime shows stronger short-timescale TKE and the summertime shows stronger long-timescale TKE. Why would relatively warm SSTs impact the 20 min - 1 hour range as opposed to a different range? Why would sea breezes and nocturnal jets contribute to TKE specifically on the 6-12 hour range? Consider citing papers in this section to justify your analysis, I don't think you necessarily need to write any code to address this point. Also, consider mentioning here that you also further investigate this line of questioning later in the paper.*

Relatively warm SSTs result in turbulence and unstable conditions. Unstable conditions are a driver for convective cells. The systems are not bound to a specific place and are advected with the wind. Therefor they can move relatively fast and can result in short timescale variability. Sea breezes on the other hand are fed by the contrast in land surface temperature and sea surface temperature. This keeps them confined to the vicinity of the intersection between land and sea, and this results in longer timescale variability.

*Fig 6: Nice figure! I think your analysis in L234-235 makes sense. I like that you have the 11 day case study to justify your hypothesis, but please include the figure somewhere, perhaps in an Appendix. Consider talking about Vincent et al (2011) citation in the preceding paragraph.*

The integrated periodogram for the 11-day case has been added to the appendix, and there have been made more references to the study of Vincent et al. (2011).

*Fig 7: You use a power curve from a 120 m turbine, but I assume you are calculating power using winds measured at 100 m, correct? You should use a turbine for which you have reliable simulated winds. If you didn't save out winds at 120 m, I believe the NREL 5 MW turbine has a 90 m hub height, and you could interpolate between your modeled winds at 80 m and 100 m. In theory you could extrapolate your modeled winds up to 120 m, but I have a feeling that would introduce a lot of noise and also uncertainty, so I recommend against that approach.*

The power curve of the Siemens SWT-3.6-120 3.6 MW turbine has been used in this case. This turbine does have a hub height of 90m (the 120 in the name is a reference to the diameter of 120 metres). We have updated the data in figure 7 and figure 8 with the interpolated 90m wind speeds, and included these figures in the paper. The difference in figure 7 is hardly noticeable (but there is a difference), for the wind speeds in figure 8 there is a small but noticeable difference. The conclusions remain the same, applying the 90m wind speeds or the 100m wind speeds.

[Figure]

(a) DJF, 6h to 12h  (b) JJA, 6h to 12h

(c) DJF, 20 min to 1h  (d) JJA, 20 min to 1h

*L270-279: I don't think that the reader was warned that this type of analysis was going to be conducted. This paragraph felt like it came out of nowhere.*

*L281-286: I don't think that the reader was warned that this type of analysis was going to be conducted. This paragraph felt like it came out of nowhere.*

A statement about this analysis has been added to the introduction of the paper.

**Minor concerns:**

1. *Figs 2 and 3: In accordance with WES "colour vision deficiency" publication guidelines, please use a different colormap than the rainbow ones.*

The colourmap for figs 2, 3 has been adjusted to matplotlibs 'viridis'. Figs 10 and 11 have also      been changed to this colourmap, as they also used the rainbow ones.

2. *L26-34: This is a suggestion and not a requirement. I found the discussion on farm density a bit hard to follow, and I wasn't certain why the authors were talking about density. Consider reorganizing the paragraph to move the thunderstorm example higher up.*

This paragraph has been reorganised.

3. *L37-38: Is there a latitude dependency for this peak? I would imagine that the timescale isn't also 4 days near the equator, but I may be wrong*

The location of this peak can change from place to place and depends on the local climate I presume. The spectrum of Kang et al. (2016) measured over Boulder for instance does not feature a on timescales longer than the diurnal cycle.

4. *L41/42: Consider citing the review papers of Stevens and Meneveau (2017) as well as the Porté-Agel et al. (2020) review paper*

These references are indeed quite monumental papers in this research area, and are added to the manuscript

5. *L96-97: You should cut this statement. If you wish to retain it, please consult the ERA5 wind energy validation that was done as part of NEWA and the Olauson (2018) paper*

This statement has been left out of the manuscript.

6. *L109: Why use RSMD instead of bias? I feel like every wind validation paper I have seen has used bias, not RSMD*

   We used the RMSD here since this metric somewhat takes the shape of both distributions into account. Two distributions with quite different means, but similar means will have a low bias.

7. *L128: You don't need to change this in the paper, and this is more for my education: do COSMO researchers talk about "periodograms"? In WRF, we call them spectra, though I suppose periodogram is more correct*

I don't have the impression that periodogram is specifically used by the COSMO community. As far as I understand a periodogram is an estimate of the underlying spectrum of a signal. For real signals it's impossible to know the underlying spectrum of a signal, due to finite time series and sample frequencies and so on. It is similar to a sample mean, which is an estimate for the full population mean. I do think that in the climate community the terms spectrum and periodogram can be used interchangeably, and I'm certain that I have called a periodogram a spectrum and vice versa. As spectra is a better known term, I guess it's better to use that term, but since I've given already too many presentations calling them periodograms I think I'll keep calling them that way.

8. *L131: Why use a window of 7 days? Could you put that into context of the mesoscale timescales you're interested in? As an aside, thank you for giving all these details on your FFTs, because people often neglect to mention these important details.*

   The 7-day window is used primarily for including (an indication of) the synoptic weather peak in the spectrum. The 1024 output intervals are not exactly equal to seven days, but in general FFT algorithms work most efficient when the length of the input vector is equal to a power of two.

9. *L147-149: If you integrate the periodogram over all bins, that's just the TKE, right? If so, maybe mention here that you take a spectral approach because you can then focus on specific scales (which would be harder to do in the time-domain)*

A statement addressing this has been added to the manuscript.

10. *L151: power curve*

This has been changed in the manuscript.

11. *L159 and 165: I recommend the authors state that "We define the MSVI..." and "We define the size of the small window...". When I read these sentences, I got the impression that some other paper specified these definitions, but I believe the MSVI is invented here.*

It has been made clearer that we constructed the MSVI metric.

12. *L191-192: This statement about the double-penalty seems very hand-wavey*

A explanation of the double penalty and why this metric might make a climate model appear       worse than it actually is, is added to the methods section. It's hard to estimate the effects of this       double penalty, and it is therefore not possible to attribute the RMSD of 1.35 m/s to this double       penalty alone. This has been added to the manuscript.

13. *L212-214: I strongly disagree with this statement. COSMO may underpredict winds in simulations without turbines, and the wakes on the lidar would conveniently also lower the observed wind speed.*

A figure plotting the availability of the Anholt wind farm has been added to the appendix. Here it can be seen that the wind farm availability over the first 100 days is much lower than over the rest of the measurement campaign. The low difference in bias and PSS between these two periods despite the large difference in availability indicates that the impact of the Anholt wind farm on the lidar is not that large. This does not mean that the Anholt wind farm has no impact on the atmosphere, but thanks to the wind blowing mainly from the west the lidar is often unwaked and this dataset is still quite useful for validation.

14. *L224-225: I appreciate that you conduct statistical testing. Is this test done to 95% confidence?*

Yes it is, and this has been added to the manuscript

15. *L289: Is an RMSD of 1.35 m/s "good agreement"? Relative to what? Either compare to other papers or reword*

This is comparable to the RMSD that for instance Wang et al. 2019 found. References to other     papers have been added to the manuscript.

16. *L298: I thought the short-timescale variability came from SST/air temperature differences, not convective systems?*

This statement is indeed a bit confusing. The gradient in the short-timescale variability implies that there is a certain spin-up period for the variability to reach its full potential, during which it is advected over the Kattegat. The variability at this timescale is therefore probably related to the unstable conditions

over sea offshore, and unstable conditions result in more convective systems. This has been clarified in the manuscript.

---

## Author Comment (AC3)

**Response to reviewer 1**

We would like to thank the reviewer for the interesting insights and constructive comments made during the critical assessment of our work. Their thorough review has greatly enriched the quality and depth of our manuscript. In what follows we address the reviewer's concerns point by point. We give a motivation to the changes, and we mention the additions to the manuscript (*in italic*).

This is a well written paper on an important topic. The paper examines the spatial and temporal variation of so-called mesoscale winds over an area with high off-shore wind energy potential. The authors show that although considering the mesoscale part of the wind speed variability doesn't change the annual power output significantly, it introduces variability into the power output. This variability is shown to peak in summer, and to have a more pronounced diurnal cycle with a daily peak at around 5pm. It is also shown to peak around coastal areas, although the authors do discuss the fact that model spin-up away from the boundaries may influence this result.

**Reply:** Thank you for pointing out the importance of this paper.

**General comments**

1. I really like the attempt to separate the 'mesoscale' and 'synoptic-scale' winds. However, I'm not really sure what physical processes are being picked up by this approach. The 'mesoscale wind', is designed to align with the effective resolution of the model, rather than any physical arguments. The 'synoptic-scale wind', calculated over a 45kmX45km grid box, is still well within the stated length-scale of 100km for mesoscale weather systems. For example, a sea-breeze disturbance could cover the whole larger box, as could a large thunderstorm complex. Can the authors try to quantify/justify this choice of length-scales? Some kind of spectral analysis or spatial filtering with known spectral properties could be helpful here. Either the length-scales should be clearly justified by physical arguments about the scale of mesoscale weather systems, or by practical arguments about the scale of variability that is of relevance to large offshore wind farms.

**Reply:** We agree with the point you bring up here and therefore added the following text to the manuscript. See also the figure hereunder which is not added to the manuscript. We also would like to mention that even though the sea-breeze indeed covers the whole box, several sea breeze episodes are picked up by the algorithm, since large variability occurs within the 45kmX45km grid box. The effective resolution is used since variability at smaller scale is not accounted for by the model.

[Figure]

©The COMET Program

*The size of the small window is determined by the effective resolution of the simulation, since variability at smaller scales is not accounted for by the model. The effective resolution of COSMO is approximately 10 × 10 grid points (Kapper et al., 2010) (~15 km). (...) With the 45x45 km² window we average out thunderstorms and smaller mesoscale systems, whereas cloud clusters and fronts are not. This is also the scale of variability that can result in sudden power ramping events for wind farms. Even though this is not strictly speaking the mesoscale length scale (100 km), we will refer to it throughout the rest of the manuscript.*

2. When discussing the variability that the authors call long time-scales (6-12 hours), the focus is on diurnal processes over the land influencing the wind over the sea. However, there can also be a diurnal cycle over the water, especially in shallow areas where the water can warm or cool more rapidly. The authors did not state how the SSTs were specified in the model, nor whether they were being updated on daily or sub-daily time-scales.

**Reply:** In the model setup section we have added the following statement regarding the SST's being provided by ERA5, and we have added a section to the appendix discussing our findings from a test run we did with a constant SST.

*The sea surface temperatures used in COSMO are inferred from the ERA5 reanalysis dataset. The SSTs do change in our simulation on a sub-daily timescale, but the diurnal cycle of these SSTs is at 1 to 2 K relatively limited. This cycle is comparable to the diurnal cycle found in potential sea surface temperature of the Baltic Sea Physics Analysis and Forecast provided by CMEMS (Lindenthal et al., 2023).*

*(Appendix): We used some test runs to get to an optimal simulation setup, and in one of these test runs the sea surface temperature (SST) was kept constant at the initial (January) level. When comparing the spectrum for both runs, we find on longer timescales no difference the intensity. On the shorter timescales there is however a higher intensity for the simulation with changing SST. As the SST does not change significantly over the daily timescale this higher intensity on short timescales might be because of the average SST in winter being higher than the SST from the first of January, thereby inducing more convective situations. In summer we find a higher intensity for the longer timescales for the simulation*

*with a constant January SST. An enhanced contrast between land and sea surface temperature might thus result in more wind speed variability on long timescales related to the land-sea breeze system.*

3. The authors looked at both temporal and spatial variability, but it would have been nice to see a greater attempt to relate these results. In particular, given that the temporal and spatial analysis should be capturing the same thing, why was it necessary to work with the spatial methods that place limitations on coastal areas? Why could the integrated periodograms over 'mesoscale' and 'synoptic-scale' periods have been compared, in a similar way to the MSVI? This would have given well-resolved maps of where the mesoscale variability was playing an important role?

**Reply:** Mesoscale systems are indeed by definition bound in both space and time, and therefore we examined mesoscale variability in both space and time. In the spatial analysis there was a limitation placed on coastal areas in order to limit the influence of the wind speed-up when transitioning from land to sea. The results of the MSVI metric become harder to interpretate when one of the windows contains a disproportionate amount of land pixels. The MSVI and the Welch method are quite complementary to each other, as one method bundles temporal information into a spectrum and the other bundles spatial information into a metric. The MSVI metric was designed to identify moments in time with strong spatial variability within an area of 45x45 km², thereby identifying the intense mesoscale systems. It therefore is not so interesting to create a map of the MSVI. The maps of the spectra integrated over different timescales have been included.

*The Welch method and the MSVI metric complement each other. The Welch method produces a spectrum with information about wind speed variations over different timescales for every pixel. Bundling this information in a spectrum does however remove the temporal resolution of that time series. The MSVI on the other hand aggregates spatial information into a metric, and in doing so gives up some spatial resolution. The temporal resolution in this method stays intact. As mesoscale systems are by definition bound in both space and time these two methods together offer a more complete view on offshore mesoscale wind speed variability than one metric would yield on its own.*

**Specific comments**

1. Page 3, line 41: "The effects of turbulence can be taken into account in Large Eddy Simulations" -> I think it should say "partly taken into account", since LES models only capture the larger part of the turbulence.

**Reply:** This is changed in the manuscript.

2. Page 3, line 53: "the majority of research is based on the onshore extent of the systems" -> There are a few references that look at the offshore part of land-sea breeze circulations that might be missing here. For example, Short et al. (2019) and Gille et al. (2005). In this context, the authors should also mention the land-breeze, which may be more relevant for offshore winds.

**Reply:** These two references are indeed very interesting and are added to the manuscript

*The offshore part of sea breezes can have an influence on the power output of a wind farm as it, in general, opposes the synoptic wind flow (Steele et al., 2015). During the night land breeze systems have the potential to generate offshore mesoscale wind speed variability (Gille et al. 2005 and Short et al. 2019).*

3. Page 2, line 38: There are more recent versions of the wind speed spectrum that you could refrence - eg. Kang et al (2016).

**Reply:** The data used in Kang et al. (2016) is taken from a continental location. For our paper we therefore opted to include a spectrum from a more coastal climate, such as Van der Hoven (1957) and Larsén et al. (2016).

4. Page 3, line 45: Change first sentence to "Less is known about the impact of mesoscale weather systems on wind variability, for example in organised convection"

**Reply:** This whole paragraph has been restructured and does no longer include the sentence: "*Less is known about mesoscale weather systems, for example in organised convection.*" Now this paragraph starts with: *"Variations in wind speed can also arise from mesoscale weather systems."*

5. Page 3, line 55: Add reference to Trombe et al. (2014).

**Reply:** This interesting paper is quite relevant and is added to the paper.

6. Page 3, lines 45-60: I think this section is missing discussion of a major source of mesoscale variability, which is from organised thunderstorms or MCS.

**Reply:** A few lines on the topic of MCSs are added to the manuscript.

*A class of convective systems that is not completely understood yet are the mesoscale convective systems (MCS) (Houze Jr, 2004). An overview of the different theories explaining the mechanisms behind MCS is summarised in for instance the introduction of Short et al. (2023). The land-sea transition can lock these MCSs in place (Xu et al., 2012), potentially affecting offshore wind farms.*

7. Page 4, line 100: What height is the ERA5 wind speed accuracy quoted for?

**Reply:** As suggested by the other reviewer, this statement is removed from the paper.

8. Page 5, line 109: 'aggregated' - clarify - is this interpolated, or averaged?

**Reply:** We have clarified in the manuscript that we mean averaged here.

9. Page 5, line 130-133: The authors state that the time series is cut in overlapping sections. Later, it says that a 'Hann window is used to cut the signal into sections'. Is this duplication?

**Reply:** This paragraph is slightly restructured to remove the duplication.

*The spectral density of a signal is estimated using a periodogram calculated with the Welch method (Welch 1967). Due to the uncertainty in a signal such as a time series of the wind speed, we can only make an estimation of the underlying spectrum. In the Welch method the 10-year time series of 10-minute interval wind speeds is cut using a Hann window in overlapping sections of approximately seven days (1024 output intervals) in our case, with an overlap of 50%. Subsequently the spectral density of every section is calculated using a Fast Fourier Transform (FFT) algorithm. The Hann window reduces the reflections arising from performing an FFT on a finite time series (Blackman and Tukey, 1958)*

**10.** Page 5, line 134: 'fast natural variability' - is it 'fast', or just removing the noise?

**Reply:** We can't really call it noise, as noise is considered an alternation to the original signal due to measurement imperfections. Our simulation should in principle not introduce something that can be classified as noise.

11. Page 6, line 135: 'good estimate' -> how do you know it's 'good'?

**Reply:** Good was meant to indicate that we consider the number of periodograms that are averaged to be large enough to converge to the underlying spectrum of wind speed variations. As this is never proven in the paper this statement of it being good is confusing, and "good" is removed from the sentence

12. Page 6, line 137: 'integrated over a 3-month interval' - what does that mean?

**Reply:** This is indeed a confusing statement so we removed it.

13. Page 6, line 144: 'period' -> can the authors choose a different word? This could mean 'a period of time' or 'periods from the spectrum'.

**Reply:** We have used "time interval" where possible to alleviate confusion for the reader.

14. Page 7, line 169: suggest changing to "As the small window is contained within the large window'

**Reply:** This is changed in the manuscript.

15. Page 7, line 170: 'everywhere' -> 'everywhere else'

**Reply:** This is changed in the manuscript.

16. Page 10, line 220: The comparison with lidar data at high frequencies is mentioned, but as far as I can see, this is not shown in the figures.

**Reply:** A reference to the figures of the comparison of the model periodogram and the lidar data in the appendix is added to the manuscript.

17. Page 10, line 225-227: Why would the diurnal effects only show up on the 12h time-scale, and not the 24-hour time-scale?

**Reply:** Masouleh et al. (2019) show that quite a large portion of sea breeze events have a duration of less than 12 hours. That is the reason why there is not one single peak in the spectrum shown in figure

5. While we believe that the temperature difference between land and sea is a driver for wind speed variability at these time scales (since the variability on these timescales is larger in summer compared to winter) the systems that this temperature contrast induces are apparently not always following this diurnal cycle.

18. Page 11, line 230: It would be useful to contrast/compare the results to Vincent (2011).

**Reply:** Indeed, the results of Vincent et al. 2011 are quite similar and a comparison is added to the manuscript.

19. Page 11, line 240: The authors mention the issues of spin-up around the edges of the domain. This is an interesting problem, but raises the question of whether the boundary removed from the edges was sufficient. Can we really trust the results, give these effects?

**Reply:** The appendix on the nesting strategy includes a small discussion on the tests we performed with a larger simulation domain. A reference to this discussion has been added to the model setup section.

*Different simulation setups have been tested for 3-month integrations. These tests included a larger domain (340 x 360 grid points compared to 180 x 184 grid points), adding an intermediate nesting at 12 km resolution and applying spectral nudging. We found no added value of using a larger domain, of adding an in between nesting step and of applying spectral nudging compared to the wind data from scatterometer. This result is in line with the findings of Ban et al. (2021) where different nesting strategies of COSMO-CLM in ERA-Interim do not show any substantial differences.*

20. Page 11, line 246: Can the authors show the periodogram for power, as well as the integrated maps? The power time-series presumably has some constant sections where the wind speed is greater than 15 m/s or less than 3 m/s, and possibly some sudden jumps due to the cut-out speed being reached. What impact do these shocks have on the periodogram?

**Reply:** These shocks elevate the spectrum quite a lot compared to the wind spectrum, but as these shocks don't necessarily happen at specific time intervals there are no clear peaks in this spectrum. The shape of the spectrum remains similar to the wind spectrum.

21. Page 14, figure 9: I don't find this graph very useful. What is it supposed to be showing? Could the authors show it as an average annual cycle, averaged over the 10-year period? Or overlay a smoothed version so that the curve is more obvious?

**Reply:** This graph is supposed to make clear to the reader that the MSVI results in a 1-D time series. This point is clarified in the text, and the figure is removed from the paper.

*The MSVI metric quantifies spatial variations in wind speed, and high MVSI values should indicate the presence of a mesoscale weather system. Indeed, looking at the wind fields associated with these peaks, a variety of mesoscale systems are clear, such as convective and sea breeze systems developing over the Kattegat.*

22. Page 15, line 269: Why are the 'mesoscale winds' always more than the 'synoptic wind speed'? This is attributed to the land-sea breeze circulation and other mesoscale phenomena. In some cases, this might not be the case, since if the sea-breeze opposes the background flow, then it will weaken the wind overall. I agree that usually, mesoscale phenomena would lead to more windy conditions, but it should not be assumed that this is the case.

**Reply:** It is true that the MSVI detects a locally elevated wind speed in each time step. Some mesoscale systems do indeed generate lower wind speeds, but due to the local nature of mesoscale weather systems a place with elevated wind speeds should be found in the vicinity of that mesoscale system. While our method is definitely not perfect the MSVI has the potential to capture this variability in wind speed created over these relatively small spatial scales.

23. Page 15, lines 270-272: This section lacks references

**Reply:** A reference for the maximum of the surface temperature in Denmark has been added to the manuscript.

*Both the mean MSVI and its diurnal amplitude is higher in summer (JJA) than winter (DJF) (fig. 10). In winter, wind speed variability is found to be more or less constant throughout the day. In summer the MSVI clearly peaks in the afternoon, when the surface temperature over land reaches its maximum (Jensen, 1960), confirming an influence of the land on the mesoscale winds over sea.*

---

## Author Comment (AC4)

**Response to reviewer 2**

We would like to thank the reviewer for the interesting insights and constructive comments made during the critical assessment of our work. Their thorough review has greatly enriched the quality and depth of our manuscript. In what follows we address the reviewer's concerns point by point. We give a motivation to the changes, and we mention the additions to the manuscript (*in italic*).

This paper conducts a 10-year mesoscale atmospheric simulation in Kattegat using the COSMO model. The goal of this paper, as stated by the authors is "to investigate what factors influence mesoscale wind speed variability, on what timescales this variability occurs, and how it affects wind power output in offshore wind farms". While I have come across many papers that conduct mesoscale wind resource assessments, I believe that a paper like this that digs deeper into the atmospheric mechanics is valuable. I really enjoyed the spectral analysis as well as the spatial analysis. I also appreciate that the authors have stored code and data on Zenodo. This is also the first mesoscale wind energy paper that I have read that doesn't use WRF, which is refreshing. That being said, I have major concerns regarding connections to the broader literature, novelty, methodology, and analysis. To the editor, I recommend a status of Major Revisions.

**Reply:** We appreciate the comments by the reviewer and are happy to hear that our efforts to dig deeper into the atmospheric processes is appreciated. A multi-model approach is very common for many research questions (impact of urbanisation, deforestation, increasing greenhouse gasses, ...) and we also see much value in doing this also for the wind energy sector. We appreciate the comments given and have done an effort to mitigate the concerns that the reviewer poses here.

**Major Concerns:**

**Framing of the paper**

To put it succinctly, I'm uncertain if the authors are claiming that either (a) few mesoscale wind simulations studies have been published or (b) if they are very specifically claiming that the mesoscale wind simulation studies that have been published have not sufficiently analyzed "mesoscale wind speed variability".

If they are claiming (a), then I strongly disagree. The authors categorize the following as mesoscale phenomena: "thunderstorms, but also sea breeze systems, low-level jets and gravity waves". Many many papers have been published on how wind energy is affected by these, e.g. (Tomaszewski and Lundquist 2020 for thunderstorms, sea breezes in the North Sea by Steele et al. (2014), many papers for LLJs, many papers by Allaerts on gravity waves). Additionally, there has been extensive work on the mesoscale through large-scale wind resource assessments (like the New European Wind Atlas and the WIND Toolkit) as well as smaller but important studies (e.g., Hahmann et al. 2015). Relatedly, I believe that NEWA, the Dutch Offshore Wind Atlas (Kalverla et al. 2020), and possibly NORA3 (Cheynet et al 2022) all include Kattegat in their simulations, and these products should be mentioned somewhere if so.

If (b), the citations in L62-63 are a small set of mesoscale wind resource papers, and I don't believe these papers do anything unique regarding wind speed variability.

**Reply:** Indeed, in retrospect we agree that the claim that "*Less is known about mesoscale weather systems...*" is not appropriate. We have restructured and rewritten large parts of the paragraph that used to start with "Less is known about mesoscale weather systems, for example in organised convection." We have added references and also mention the wind atlases. We hope that rewriting part of the introduction results in a better framing for the paper. The novelty of the paper is addressed in the reply to the next point.

*Variations in wind speed can also arise from mesoscale weather systems. With their length scales ranging up to a hundred kilometres and timescales spanning from ten minutes to a few hours, mesoscale weather systems occupy an intermediary position between turbulence and synoptic weather systems. (...) Wind atlases utilising these models at a kilometre grid have been made available to the public, such as the Dutch Offshore Wind Atlas (DOWA) (Wijnant et al., 2019) and the New European Wind Atlas (NEWA) (Petersen et al., 2014; Hahmann et al., 2020; Dörenkämper et al., 2020), and provide thoroughly validated information at the mesoscale level (Kalverla et al., 2020). Both Hahmann et al. (2015) and Wang et al. (2019) have evaluated kilometre scale climate models against in situ data and have shown the capacity of these models to reproduce mesoscale variability. (...)*

**Novelty**

I normally don't comment on novelty, but many mesoscale simulations have been conducted specifically in the North Sea, with domains that include Kattegat. A non-exhaustive list includes Hahmann et al. (2015) and the New European Wind Atlas (see the two papers in GMD). I believe that the Dutch Offshore Wind Atlas (see papers by Kalverla) and possibly NORA3 (Cheynet et al. 2022) may also cover this region. I don't see any references to these papers. The authors should clearly state why their work is novel to help the readers out. I feel like I've seen many papers also do similar-but-different seasonal analysis before (e.g., Wang et al 2019 in California), and these should also at least be referenced.

**Reply:** We included several lines on the novelty of our study in the introduction, and we included the references mentioned above.

*In our paper we use a 10-year integration of the convection permitting climate model COSMO-CLM with a horizontal resolution of 1.5 km to study the atmospheric mechanics behind mesoscale variability. Augmenting existing research, we added here the implications for power production for a full 10-year period. Moreover, this paper introduces a new index, which identifies spatially coherent mesoscale systems, allowing for detection of situations with a strong mesoscale system. We add on this spatial index with temporal analysis methods, since these methods are inherently complementarity. Additionally, this paper uses COSMO-CLM, which complements the frequently used model WRF, enabling a multi-model approach to applications in the wind energy sector in the future. Our analysis has been performed using 10-minute wind speed data, which is not available from wind atlases. (...)*

**Methodology**

Simulation setup: What is the timestep of the simulation? How frequently is data saved out? Did you run a single, 10-year long simulation or did you chunk up the job into smaller periods? Did you use spectral nudging to encourage the long simulation to not drift too far away from the expected ERA5 values? I'm unfamiliar with COSMO, but in WRF, it is known that the PBL scheme choice is very important. If multiple options are allowed in COSMO, which PBL scheme is used here? While not required by this

journal, I highly encourage the authors to upload an example configuration file somewhere so that others may more easily reproduce this study in the future. There is possibly one on Zenodo, but the link is private, so I cannot view its contents.

**Reply:** We added this information to the manuscript, and we added a configuration file to the Zenodo link. Additional information about the nesting strategy is mentioned in appendix A. Except for the 10, 80 and 100 m wind speed which are saved out every ten minutes, all variables are saved out hourly.

*The dynamical core of this model solves the primitive thermo-hydrodynamical equations describing a compressible flow in a moist atmosphere (Doms and Baldauf, 2018) with a timestep of 10 seconds. (...) Subgrid-scale turbulence is parametrised by a one-dimensional diagnostic level 2.5 closure scheme based on a prognostic TKE equation (Raschendorfer, 2001; Schulz, 2008a, b). (...) For practical reasons the calculation of the 10-year simulation has been divided into smaller periods, and by using the restart files generated by COSMO these periods were initialised with a warm start. (...)*

*Different simulation setups have been tested for 3-month integrations. These tests included a larger domain (340 × 360 grid points compared to 180 × 184 grid points), adding an intermediate nesting at ≈ 12 km resolution and applying spectral nudging. We found no added value of using a larger domain, of adding an in between nesting step and of applying spectral nudging compared to the wind data from scatterometer. This result is in line with the findings of Ban et al. (2021) where different nesting strategies of COSMO-CLM in ERA-Interim do not show any substantial differences.*

Validation efforts: Researchers often validate modeled winds against measured winds in order to built trust, but the validation study here instead erodes my confidence. I also don't see how validation furthers the authors stated goals in the intro. I believe that all the lidar analysis should be struck entirely. Please write a sentence or two that explicitly ties the motivation behind the validation back to the goal of the paper.

**Reply:** The validation with lidar data is included in the paper in order to validate the wind speeds at 100 metres, as we mainly use these wind speeds in the paper. We added these lines to the introduction to motivate the validation efforts. We do not agree that the lidar comparison should be removed from the manuscript (see point below).*The goals mentioned above can only be achieved if the model represents the real atmospheric winds. Therefore, we have incorporated an evaluation of the near-surface winds using ASCAT data (ref) and the 100-m winds using lidar data. These nicely complement each other as the ASCAT measures multiple locations, but only at 9h UTC and 21h UTC, while the lidar data provides measurements during the whole day, albeit at one single location.*

Lidar: The simulated winds do not include wakes, but the lidar is probably being waked. That waking can be significant (>1 m/s modifications). We also don't know how close the lidar is to the nearest turbine, and also the number of operational turbines changes throughout the comparison period, changing the waking strength. The authors attempt to minimize the effects of waking by looking at periods when wind farm availability stayed below 50%, but that is insufficient in my opinion.

**Reply:** In response to the editor's comments in an earlier stage about the fact that only near surface winds were initially evaluated but not the hub height wind speeds, the comparison with lidar was added. We agree that it is not an ideal location, but it is the only offshore data that are available. In addition, the fact that there is not a large difference in performance between the first 100 days and the entire

period gives us confidence of the added value of including the lidar data (see the wind farm availability figure added hereunder). We agree that the reader should be aware of the caveats and therefore explicitly state the ones that you mentioned in the manuscript now.

[Figure]

*Using scatterometer data only the near surface wind speeds at 9h and 21h UTC of our model can be evaluated. As a complementary validation of the scatterometer a light detection and ranging (lidar) device located 2 kilometres west of the Anholt wind farm is used. This is not an ideal location, since it might be affected by the wind farm which is not implemented in our model, but it is the only offshore measurement at hub height in the Kattegat area. Therefore, a sub period of the measurement campaign was used, one in which the Anholt wind farm was still under construction and the availability of operation wind farms increased from xx to xx%. Note that eventually xx windfarms were operational in the Anholt wind farm. Moreover, the wind mainly blows from the west in this area (Karagali et al., 2013), resulting in a lidar signal that is largely unwaked. (...)*

ASCAT: I am less familiar with this instrument, but I suggest that the authors point out that others in wind energy have used this data source before (e.g., Hasager et al. 2020). I mention this because I know there is some controversy regarding validating WRF against SAR (a different instrument), but if others in the wind energy sector have looked at ASCAT previously, then there is a stronger case for its use here.

**Reply:** A reference to the validation of the DOWA with ASCAT has been added to the paper, as well as a reference to Hasager et al. 2020.

*Validation with ASCAT data has already been used for a variety of offshore wind datasets (Hasager et al., 2020; Duncan et al., 2019).*

ASCAT: I haven't seen others make maps of winds at different percentiles. Why did you do this instead of simply comparing mean wind speed maps? You give a justification on L110, but I don't quite follow. Maybe if you provide a summary of what the "double penalty" is, that would help clarify things.

**Reply:** We have added an explanation of how a double penalty could impact the comparison between scatterometer data and simulation output.

*A direct point-to-point comparison between ASCAT and model output, like RMSD, might underestimate the quality of the model to represent the mesoscale variability. In the absence of strong forcing over sea, it can easily happen for a model to reproduce a mesoscale system, albeit slightly shifted in time and/or space. The reproduced mesoscale system then results in two errors. First it induces an error over the place where it should have been but is not reproduced now, and secondly it induces an error over the place where it is now but should not have been, which is referred to as the double penalty (Marseille and Stoffelen, 2017). That is why, apart from the RMSD, statistical methods to assess the distribution of wind speeds, like the 25th, 50th and 75th percentile wind speeds, are also used. The comparison of these distribution parameters has an added value compared to an evaluation of the mean, as an erroneous distribution can still have a good representation of the mean due to error compensation. Evaluating distribution parameters is therefore more rigorous than evaluating only the mean.*

ASCAT: L106: It's hard to judge if 229,503 WVC is a lot of data or a small amount of data. Could you give more context? Maybe an easy number to calculate is the number of aggregated WRF grid cells over this period.

**Reply:** We have added a calculation of the number of aggregated simulation grid cells to the manuscript.

*The scatterometer does not cover the Kattegat on every overpass it makes, yet a total of $299,503$ Wind Vector Cells (WVC), which is equivalent to ≈ $20,800,000$ aggregated (i.e. averaged) model grid cells, is available for comparison.*

ASCAT: If a model validates well at 10 m, that doesn't necessarily mean that the model is accurate at hub-height (see the Bodini extrapolation papers if you're interested). Please add that caveat in somewhere, as many today would consider validating against near-surface winds to be of minimal utility (for what it's worth, I am not one of those people).

**Reply:** We have added this caveat to the manuscript.

*Using scatterometer data only the near surface wind speeds at 9h and 21h UTC of our model can be evaluated, which is not necessarily representative for the hub-height wind speed. As a complementary validation of the scatterometer a light detection and ranging (lidar) device located 2 kilometres west of the Anholt wind farm is used.*

**Analysis**

Fig 5: I really like the spectral analysis. One of the goals of this paper is to "to investigate what factors influence mesoscale wind speed variability", and as such, I would like to see a stronger physical justification as to why the wintertime shows stronger short-timescale TKE and the summertime shows stronger long-timescale TKE. Why would relatively warm SSTs impact the 20 min - 1 hour range as opposed to a different range? Why would sea breezes and nocturnal jets contribute to TKE specifically on the 6-12 hour range? Consider citing papers in this section to justify your analysis, I don't think you necessarily need to write any code to address this point. Also, consider mentioning here that you also further investigate this line of questioning later in the paper.

**Reply:** We added a more physical explanation of how winter conditions favour short timescale variability, and summer conditions result mainly in long timescale variability to the manuscript.

*Relatively warm SSTs in winter result in turbulence and unstable conditions. Unstable conditions are a driver for convective cells. The systems are not bound to a specific place and are advected with the wind. Therefore they can move relatively fast and result in short timescale variability (20min-1h) (Ahrens, 1994). (...) The difference between DJF and JJA on the longer timescales may be due to the sun being higher in summer, and it heating the land more effectively. With the sun over land, the air above it expands and generates a breeze over the sea during the morning or afternoon. During the night, due to the land cooling nocturnal jets may be formed. Sea breezes are fed by the contrast in land surface temperature and sea surface temperature. This keeps them confined to the vicinity of the intersection between land and sea, and this results in longer timescale variability (Ahrens, 1994).*

Fig 6: Nice figure! I think your analysis in L234-235 makes sense. I like that you have the 11 day case study to justify your hypothesis, but please include the figure somewhere, perhaps in an Appendix. Consider talking about Vincent et al (2011) citation in the preceding paragraph.

**Reply:** The integrated periodogram for the 11-day case has been added to the appendix, and there have been made more references to the study of Vincent et al. (2011).

Fig 7: You use a power curve from a 120 m turbine, but I assume you are calculating power using winds measured at 100 m, correct? You should use a turbine for which you have reliable simulated winds. If you didn't save out winds at 120 m, I believe the NREL 5 MW turbine has a 90 m hub height, and you could interpolate between your modeled winds at 80 m and 100 m. In theory you could extrapolate your modeled winds up to 120 m, but I have a feeling that would introduce a lot of noise and also uncertainty, so I recommend against that approach.

**Reply:** The power curve of the Siemens SWT-3.6-120 3.6 MW turbine has been used in this case. This turbine does have a hub height of 90m (the 120 in the name is a reference to the diameter of 120 metres). We have updated the data in figure 7 and figure 8 with the interpolated 90m wind speeds and included these figures in the paper. The difference in figure 7 is hardly noticeable (but there is a difference), for the wind speeds in figure 8 there is a small but noticeable difference. The conclusions remain the same, applying the 90m wind speeds or the 100m wind speeds.

*Periodograms can also be used to examine the fluctuations in potential wind power. The power curve of a wind turbine converts a wind speed time series to a wind power time series, and of the latter time series a periodogram can be calculated. The power curve of the Siemens SWT-3.6-120 wind turbine is available in table form (Bauer and Matysik, 2022) and a cubic spline interpolation is used to obtain the power for every possible wind speed. As the hub height for this type of turbine is 90 metres the 80 metre and 100 metre model wind speeds are interpolated to 90 metres via the power law given in equation 1.*

[Figure]

(a) DJF, 6h to 12h     (b) JJA, 6h to 12h

(c) DJF, 20 min to 1h     (d) JJA, 20 min to 1h

L270-279: I don't think that the reader was warned that this type of analysis was going to be conducted. This paragraph felt like it came out of nowhere.

L281-286: I don't think that the reader was warned that this type of analysis was going to be conducted. This paragraph felt like it came out of nowhere.

**Reply:** These analyses are now introduced in the introduction and explained in the methods section of the manuscript.

*The added value of a convection permitting simulation over ERA5 for wind and power resources is assessed in this section.*

*A comparison between ERA5 and our simulation output was performed using the MSVI, assessing the added value of convection permitting simulations for mesoscale variability. This is done by regridding ERA5 to the grid of our COSMO simulation output, and then applying the MSVI metric to the ERA5 data.*

*Given the offshore wind speed variability, wind power fluctuations are expected. Wind speed and wind power are related to each other via the power curve of a wind turbine, and given this relation an MSVI analysis on the wind power could in principle be made. Yet the shape of this power curve imposes restrictions on our methodology: below the cut-in wind speed of a turbine (3 m/s) no power is produced, prohibiting a MSVI calculation for power fluctuations as the denominator would become zero. Instead we opt for a RMSD comparison between power time series for a stationary 10 x 10 window and a 30 x 30 window. The stationary windows are positioned in the area of the Anholt wind farm. The small window is*

*in area nine times smaller than the large window. In order to cover the whole large window, nine small window power time series are calculated and compared to the large window power time series. The average over these nine RMSD values is then taken to assess the differences in wind power.*

**Minor concerns:**

1. Figs 2 and 3: In accordance with WES "colour vision deficiency" publication guidelines, please use a different colormap than the rainbow ones.

**Reply:** The colourmap for figs 2, 3 has been adjusted to matplotlibs 'viridis'. Figs 10 and 11 have also been changed to this colourmap, as they also used the rainbow ones.

2. L26-34: This is a suggestion and not a requirement. I found the discussion on farm density a bit hard to follow, and I wasn't certain why the authors were talking about density. Consider reorganizing the paragraph to move the thunderstorm example higher up.

**Reply:** This paragraph has been reorganised.

3. L37-38: Is there a latitude dependency for this peak? I would imagine that the timescale isn't also 4 days near the equator, but I may be wrong

**Reply:** The location of this peak can change from place to place and depends on the local climate I presume. The spectrum of Kang et al. (2016) measured over Boulder for instance does not feature a on timescales longer than the diurnal cycle.

4. L41/42: Consider citing the review papers of Stevens and Meneveau (2017) as well as the Porté-Agel et al. (2020) review paper

**Reply:** These references are indeed quite monumental papers in this research area, and are added to the manuscript.

5. L96-97: You should cut this statement. If you wish to retain it, please consult the ERA5 wind energy validation that was done as part of NEWA and the Olauson (2018) paper

**Reply:** This statement has been left out of the manuscript.

6. L109: Why use RSMD instead of bias? I feel like every wind validation paper I have seen has used bias, not RSMD

**Reply:** We used the RMSD here since this metric somewhat takes the shape of both distributions into account. Two distributions with quite different means, but similar means will have a low bias.

7. L128: You don't need to change this in the paper, and this is more for my education: do COSMO researchers talk about "periodograms"? In WRF, we call them spectra, though I suppose periodogram is more correct

**Reply:** I don't have the impression that periodogram is specifically used by the COSMO community. As far as I understand a periodogram is an estimate of the underlying spectrum of a signal. For real signals

it's impossible to know the underlying spectrum of a signal, due to finite time series and sample frequencies and so on. It is similar to a sample mean, which is an estimate for the full population mean. I therefore think the term periodogram is better. I do think that in the climate community the terms spectrum and periodogram can be used interchangeably, so we added the following line to the manuscript.

*The spectral density of a signal is estimated using a periodogram, also referenced to as a spectrum, calculated with the Welch method.*

8. L131: Why use a window of 7 days? Could you put that into context of the mesoscale timescales you're interested in? As an aside, thank you for giving all these details on your FFTs, because people often neglect to mention these important details.

**Reply:** The 7-day window is used primarily for including (an indication of) the synoptic weather peak in the spectrum. The 1024 output intervals are not exactly equal to seven days, but in general FFT algorithms work most efficient when the length of the input vector is equal to a power of two.

*In the Welch method the 10-year time series of 10-minute interval wind speeds is cut using a Hann window in overlapping sections of approximately seven days (1024 output intervals) in our case, with an overlap of 50%. This window length allows for part of the synoptic peak in periodogram to be seen.*

9. L147-149: If you integrate the periodogram over all bins, that's just the TKE, right? If so, maybe mention here that you take a spectral approach because you can then focus on specific scales (which would be harder to do in the time-domain)

**Reply:** A statement addressing this has been added to the manuscript.

*These averaged periodograms can be integrated over a time interval of interest resulting in one single value per grid point that quantifies the temporal variability over that time interval. This makes it possible to visually compare the different grid points for each season and allows us to focus on specific scales, which would be challenging in the time-domain.*

10. L151: power curve

**Reply:** This has been changed in the manuscript.

11. L159 and 165: I recommend the authors state that "We define the MSVI..." and "We define the size of the small window...". When I read these sentences, I got the impression that some other paper specified these definitions, but I believe the MSVI is invented here.

**Reply:** It has been made clearer that we constructed the MSVI metric.

12. L191-192: This statement about the double-penalty seems very hand-wavey

**Reply:** An explanation of the double penalty and why this metric might make a climate model appear worse than it actually is, is added to the methods section. It's hard to estimate the effects of this double

penalty, and it is therefore not possible to attribute the RMSD of 1.35 m/s to this double penalty alone. This has been added to the manuscript.

*Moreover, the 1.35 m/s RMSD may be explained by the double penalty mentioned earlier, but there is no straightforward way of testing this. However, tests with spectral nudging did not substantially improve the performance indicating that the lateral boundaries to a large extent control the timing and location of weather systems for this domain and model configuration.*

13. L212-214: I strongly disagree with this statement. COSMO may underpredict winds in simulations without turbines, and the wakes on the lidar would conveniently also lower the observed wind speed.

**Reply:** A figure plotting the availability of the Anholt wind farm has been added to the appendix. Here it can be seen that the wind farm availability over the first 100 days is much lower than over the rest of the measurement campaign. The low difference in bias and PSS between these two periods despite the large difference in availability indicates that the impact of the Anholt wind farm on the lidar is not that large. This does not mean that the Anholt wind farm has no impact on the atmosphere, but thanks to the wind blowing mainly from the west the lidar is often unwaked and this dataset is still quite useful for validation. A few lines on this have been added to the results of our manuscript.

*Even though there is some uncertainty about the effect of the wind farm on the lidar data, the small difference in performance between the two periods, together with the close correspondence between lidar data and COSMO gives us some confidence that the model performance is adequate. It might however be possible that COSMO under predicts winds in the simulation without turbines, which is masked by the wind farm wakes experienced by the lidar. However, the effect is likely small, due to the wind mainly blowing from the west in this area (Karagali et al., 2013).*

14. L224-225: I appreciate that you conduct statistical testing. Is this test done to 95% confidence?

**Reply:** Yes it is, and this has been added to the manuscript.

*Using a Student's t-test we find that the differences over these time slots are significant at the 0.05 level.*

15. L289: Is an RMSD of 1.35 m/s "good agreement"? Relative to what? Either compare to other papers or reword

**Reply:** This is comparable to the RMSD that for instance Wang et al. 2019 found.

*The simulation showed good agreement with scatterometer observations away from coasts and small islands with a spatiotemporal root mean square difference of 1.35~m/s, which is comparable to for instance Wang et al. (2019) found.*

16. L298: I thought the short-timescale variability came from SST/air temperature differences, not convective systems?

This statement is indeed a bit confusing. The gradient in the short-timescale variability implies that there is a certain spin-up period for the variability to reach its full potential, during which it is advected over

the Kattegat. The variability at this timescale is therefore probably related to the unstable conditions over sea offshore, and unstable conditions result in more convective systems. This has been clarified in the manuscript.

*Our results show that more variability in wind speed is expected in winter due to unstable conditions over sea. These unstable offshore conditions result in increased turbulence and induce convective systems, which generate wind speed variability on short timescales (20 minutes to 1 hour).*

---

## Author Response (AR2)

**Response to reviewer1**

1. I do find it difficult to understand that the metric is being designed based on physical considerations, rather than on the practical considerations resulting from the grid-spacing of the model. Perhaps you could rename this the 'Multi-Scale Variability Index', and avoid this problem of claiming that it separates the mesoscale and synoptic scales? The relationship of this to the mesoscale and synoptic scale variability could still be discussed, but this would also make the methodology quite flexible for others to apply in other scenarios where it is of interest to separate the variability from two time-scales.

**Reply:** We added a paragraph to the manuscript to explain better our approach. Although such a definition always relies on a combination of physical and practical considerations we prefer not to use the term multiscale spatial variability index, since this term does not explicitly refer to the scale it aims to capture.

*One limitation is that this metric is derived using a model run of 1.5 km grid spacing, which is a slightly higher resolution than most wind atlases (e.g. DOWA Wijnant et al. (2019) and NEWA Petersen et al. (2014); Hahmann et al. (2020); Dörenkämper et al. (2020)), so the smallest signals that can be captured are at the effective resolution of 10×10 grid points (Kapper et al., 2010) (∼15 km). As size for the small window this effective resolution is chosen, thereby capturing most of the mesoscale variability, although the smallest mesoscale signals are averaged out by this or any other product based on state-of-the-art mesoscale model simulations. The synoptic scale starts at ∼100 kilometres (Oblack, 2020), but when taking this size for the large window, frontal systems were averaged out. Therefore, we took approximately half of this as dimensions for the large window to fully capture the synoptic background velocity field including the frontal systems. A size of 30 grid points (∼45 km) is chosen for the large window size, which is in area nine times larger than the small window. With these window sizes, the MSVI turned out to identify mesoscale systems like gravity waves, convective systems and land-sea breezes (see Section 3.3).*

2. Line 220-231: I appreciate the added attempt to explain the window sizes. There is a grammatical problem in the added text: "we average out thunderstorms and smaller mesoscale systems, whereas cloud clusters and fronts are not". I suggest changing this to "thunderstorms and smaller mesoscale systems are averaged out, whereas cloud clusters are not".

**Reply:** This has been changed in the manuscript.

*Using the large window thunderstorms and smaller mesoscale systems are averaged out, whereas cloud clusters and fronts are not.*

3. Line 64: "The offshore part of sea breezes can have an influence on the power output of a wind farm as it, in general, opposes the synoptic wind flow (Steele et al., 2015)." -> I don't think there is anything inherent about the offshore part of sea breezes opposing the synoptic wind flow. This would be sites-specific, and depend on the orientation of the coastline and the prevailing background wind. Also, are the authors talking about the offshore part of the Seabreeze, or the landbreeze?

**Reply:** This has been adjusted in the manuscript.

*The offshore part of pure sea breezes can have an influence on the power output of a wind farm as it, in general, opposes or reinforces the synoptic wind flow (Steele et al., 2015).*

4. Line 67: What aspect of MCSs is not understood? I agree this is a complex topic, but it would be helpful to understand the relevant of this sentence to this study.

**Reply:** This has been clarified in the manuscript.

*A class of convective systems of which the conditions for formation, evolution and stability are not completely understood yet are the mesoscale convective systems (MCS) (Houze Jr, 2004).*

5. Line 280: ..."meaning that parametrising the effects of the wind farm with Fitch wind introduce a larger error than not taking it into account". If the Fitch parametrisation is going to be mentioned here, then it needs to be introduced and referenced properly. I also don't understand the connection between the first and second parts of this sentence.

**Reply:** In response to a comment from the other reviewer this statement has been cut from the manuscript.

6. Line 296: I don't believe the Van der Hoven (1957) spectrum is an appropriate reference for this paper. This work was based on an extremely limited dataset of only a few hours for the higher frequencies, and the higher end of the spectrum was measured during a tropical cyclone, which is not relevant to this study.

**Reply:** We refer to a more modern measurement of the atmospheric spectrum.

*For the lower frequencies the periodogram seems to be leveling off, which is indicative of the synoptic weather peak (Larsén et al., 2016).*

**Response to reviewer 2**

1. L77-78: NREL's WIND Toolkit is available at 5 min resolution, so adjust the sentence to say something like "which is not available from wind atlases in our region of interest"

**Reply:** This has been changed in the manuscript.

*Our analysis has been performed using 10-minute wind speed data, which is not available from wind atlases in our region of interest.*

2. While I still believe that the lidar measurements should be excluded from this study, I will not press this larger issue further, and I appreciate that the authors have added more caveats. I have also now seen the editor's initial concerns, and I can see additional added value in doing lidar analysis in this context. That being said, I do request some small tweaks regarding the words used around the lidar analysis:
   a. L149: Does availability rise from 12 to 36%, or normalized availability rise from 12% to 36%? Fig A3 suggests normalized availability.

**Reply:** In the manuscript this has been changed to the latter.

*Therefore a sub period of the measurement campaign was used, one in which the Anholt wind farm was still under construction and the normalised availability of operation wind farms increased from 12% to 36% (see fig. A3 for the availability of the Anholt wind farm).*

   b. L154: I'm being pedantic, but it's only "appropriate" to compare time averages and spatial averages if we know that the data is ergodic, and this data probably isn't strictly ergodic. Could you soften this statement to say "This is complementary" or something along those lines?

**Reply:** We have softened this statement in the manuscript.

*Note that the COSMO wind speed values are a grid cell average. This spatial averaging, similar to the temporal averaging of the lidar data, should diminish the impact of wind gusts on the analysis.*

   c. L264-265: I'm not convinced of this statement. If the winds were weak during this period, then Fitch would have handled the low availability just fine. Please cut the "During this... into account." Statement.

**Reply:** This statement has been cut from the manuscript.